# CHARACTERIZING ROBUST OVERFITTING IN ADVERSARIAL TRAINING VIA CROSS-CLASS FEATURES

## ABSTRACT

Adversarial training (AT) has been considered one of the most effective methods for making deep neural networks robust to adversarial attacks. However, AT can lead to a phenomenon known as robust overfitting where the test robust error gradually increases during training, resulting in a large robust generalization gap. In this paper, we present a novel interpretation of robust overfitting from the perspective of feature attribution. We find that at the best checkpoint of AT, the model tends to involve more cross-class features, which are shared by multiple classes, in its decision-making process. These features are useful for robust classification. However, as AT further squeezes the training robust loss, the model tends to make decisions based on more class-specific features, giving rise to robust overfitting. We also provide theoretical evidence for this understanding using a synthetic data model. In addition, our understanding can also justify why knowledge distillation is helpful for mitigating robust overfitting, and we further propose a weight-average guided knowledge distillation AT approach for improved robustness.

## 1 INTRODUCTION

As the existence of adversarial examples (Goodfellow et al., 2014) has led to significant safety concerns of deep neural networks (DNNs), a series of methods (Papernot et al., 2016; Cohen et al., 2019; Chen et al., 2023) for defending against this threat have been proposed. Adversarial training (AT) (Madry et al., 2017), which adds adversarial perturbations to samples in the training loop and encourages the model to distinguish these perturbed samples, has been considered one of the most effective ways to make the DNNs more robust to adversarial attacks (Athalye et al., 2018). AT can be formulated as the following min-max optimization problem:

$$\min_{\boldsymbol{\theta}} \mathcal{L}(\boldsymbol{\theta}), \quad \text{where} \quad \mathcal{L}(\boldsymbol{\theta}) = \frac{1}{N} \sum_{i=1}^{N} \max_{\|\delta_i\|_p \leq \epsilon} \ell(f(\boldsymbol{\theta}, x_i + \delta_i), y_i), \tag{1}$$

where $\boldsymbol{\theta}$ represents the model parameter, $\ell$ is the loss function (such as cross-entropy loss), $(x_i, y_i)$ is the $i$-th sample-label pair in the training set for $1 \leq i \leq N$, and $\epsilon$ is the perturbation bound.

Despite the success in improving adversarial robustness, AT can also lead to a phenomenon known as *robust overfitting* (Rice et al., 2020). During AT, a model may achieve its best test robust error at a certain epoch, but the test robust error will gradually increase in the latter stage of training. By contrast, the training robust error consistently decreases, resulting in a large robust generalization gap. As robust overfitting exposes a fundamental limitation in AT, several techniques have been introduced to address this issue, such as knowledge distillation (Chen et al., 2021). However, there is still a lack of complete understanding regarding the underlying mechanism of how such robust overfitting occurs.

In this paper, we characterize the phenomenon of robust overfitting from the perspective of feature attribution. Specifically, we divide the features learned by the model into *cross-class* features and *class-specific* features. The cross-class features are shared among multiple classes in the classification task, *e.g.* the feature *wheels* shared by the *automobile* and *truck* classes in the CIFAR-10 dataset. We investigate how these features are used in the decision-making process of the model in AT. Intriguingly, we observe that at the best checkpoint during AT, the model relies more on cross-class

features than at later checkpoints. In contrast, at later checkpoints where robust overfitting occurs, the model tends to make decisions based on more class-specific features that are specified to only one class.

Motivated by this observation, we propose a novel interpretation of robust overfitting. During the initial stage of AT, the model learns both class-specific and cross-class features simultaneously, since these features are both helpful for reducing robust loss when this loss is large. However, as training progresses and the robust loss decreases to a certain degree, the model begins to abandon cross-class features and makes decisions based mainly on class-specific features. This is because cross-class features raise positive logits on other classes and yield non-zero robust loss in AT. Therefore, the model tends to abandon these features to further decrease the robust loss. However, these cross-class features are helpful for robust classification (*e.g.*, a feature shared by classes $y_1, y_2$ helps the model distinguish samples from class $y_1$ to other classes $y_3, \cdots, y_n$), and using only class-specific features is insufficient to achieve the best robust accuracy. This results in a decline in robust test accuracy and leads to robust overfitting.

We provide both empirical and theoretical evidence to support this interpretation. First, we propose a metric to characterize the usage of the cross-class features for a certain model. Then, among different perturbation norms, datasets, and architectures, we show that the overfitted models consistently tend to use fewer cross-class features. We further provide theoretical evidence to support this understanding using a synthetic dataset that decouples cross-class and class-specific features. In our theoretical framework, we show that cross-class features are more sensitive to robust loss, but they are indeed helpful for robust classification.

In addition, our understanding can justify how knowledge distillation helps alleviate robust overfitting (Chen et al., 2021) by showing that knowledge distillation can preserve cross-class features during AT. Furthermore, we aim to introduce a better teacher model to characterize more precise cross-class features. Motivated by the fact that weight averaging can improve robustness in AT (Wang & Wang, 2022), we propose utilizing such a model as the teacher model for better knowledge distillation in AT. Experiment demonstrates that our approach exhibits better robustness performance than previous approaches.

Our contributions can be summarized as follows:

1. We propose a novel interpretation of robust overfitting in AT. We show that a key factor of robust overfitting is that in order to achieve lower robust loss, the model tends to reduce the reliance on cross-class features, which are actually helpful for robust classification.

2. We provide both empirical and theoretical evidence to support our proposed understanding. Empirically, we illustrate that overfitted models in AT use fewer cross-class features than the best checkpoints. We also substantiate these assertions in a synthetic data model with decoupled cross-class and class-specific features.

3. Our understanding also shows that knowledge distillation helps mitigate robust overfitting by preserving these features. Considering weight-averaged models can provide better information on cross-class features, we propose to use such models for knowledge distillation in AT for improved robustness.

## 2 BACKGROUND AND RELATED WORK

### 2.1 ADVERSARIAL TRAINING AND ROBUST OVERFITTING

Adversarial training (AT) has been widely recognized as one of the most effective approaches to improving the robustness of models. The optimization objective of AT is shown in equation (1). For the inner maximization, Projected Gradient Descent (PGD) is generally used to craft the adversarial example:

$$x^{t+1} = \Pi_{\mathcal{B}(x,\epsilon)}(x^t + \alpha \cdot \text{sign}(\nabla_x \ell(\theta; x^t, y))), \tag{2}$$

where $\Pi$ is the function that projects the sample onto an allowed region of perturbation, *i.e.*, $\mathcal{B}(x, \epsilon) = \{x' : \|x' - x\|_p \leq \epsilon\}$, and $\alpha$ controls the step size of gradient ascent. However, AT suffers from the problem of robust overfitting (Rice et al., 2020). As shown in Figure 1, the model may perform best on the test dataset at a certain epoch during AT, but in the later stages, the model's performance on

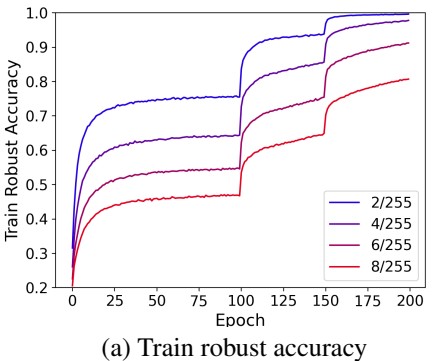
(a) Train robust accuracy

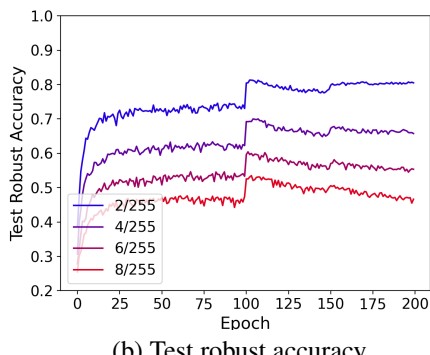
(b) Test robust accuracy

Figure 1: Train and test robust accuracy of AT on CIFAR-10 dataset with $\ell_\infty$-norm perturbation bound $\epsilon \in \{2/255, 4/255, 6/255, 8/255\}$.

the test data gradually worsens. Meanwhile, the model's robust error on the training data continues to decrease, leading to a significant generalization gap in adversarial training. Moreover, the commonly used perturbation bound $\epsilon$ (*e.g.* $[0, 8/255]$ for $\ell_\infty$-norm) in AT, a relatively large $\epsilon$ suffers from more severe robust overfitting. By contrast, for a small $\epsilon = 2/255$, this effect is relatively less pronounced.

## 2.2 UNDERSTANDING AND ALLEVIATING ROBUST OVERFITTING

To address the robust overfitting issue in AT, several techniques have been introduced from various perspectives. For example, introducing low curvature activation (Singla et al., 2021), data augmentation (Rebuffi et al., 2021b; Li & Spratling, 2023) and temporal ensembling (Dong et al., 2022) are helpful to mitigate robust overfitting. One series of works attempted to understand and alleviate this overfitting by attributing robust overfitting to the sharpness of the weight loss landscape (Li et al., 2018) and propose to introduce flatness as a regularization (Wu et al., 2020; Yu et al., 2022) to mitigate this effect. Another representative method is injecting *smoothening* during AT (Chen et al., 2021), which introduces knowledge distillation (Hinton et al., 2015) in AT to smooth the logits and leverage stochastic weight averaging (SWA) (Izmailov et al., 2018) to smooth the weights. The loss function of AT with knowledge distillation can be formulated as

$\min_\theta \mathbb{E}_{(x,y)\sim\mathcal{D}_{train}} \left[ \max_{\|\delta\|_p \le \epsilon} \tilde{\ell}(\theta; \theta_1, \theta_2, x+\delta, y) \right]$, where

$$\tilde{\ell}(\theta\,;\,\theta_1, \theta_2, x+\delta, y) = (1-\lambda_1-\lambda_2)\ell_{CE}(f(\theta, x+\delta), y) + \sum_{i=1}^{2} \lambda_i\,\mathcal{KD}(f(\theta, x+\delta), f(\theta_i, x+\delta)) \quad (3)$$

where $\ell_{\text{CE}}$ is the cross-entropy loss, and $\mathcal{KD}$ is the knowledge distillation function (details in Appendix G.2), and $\theta_1$ and $\theta_2$ are the robust-/standard-trained *self-teachers*, respectively. SWA can be expressed as

$$\theta_{\text{SWA}}^T = \frac{n\,\theta_{\text{SWA}}^{T-1} + \theta^T}{n+1}, \tag{4}$$

where $T$ is the current training epoch, $n$ is the number of checkpoints involved in weight averaging and $\theta_{\text{SWA}}$ represents the averaged model parameter. While these methods have been proven useful in mitigating robust overfitting, there is still a lack of comprehensive understanding of the underlying mechanisms of how robust overfitting occurs and why knowledge distillation is useful in mitigating it.

## 3 PROPOSED UNDERSTANDING

In this section, we elaborate on our proposed understanding of robust overfitting in AT via cross-class features. We first present a metric of cross-class feature usage for a model in AT. Then, with comprehensive empirical evidence, we demonstrate how robust overfitting occurs based on the dynamics of the model learns and abandons these features during AT.

## 3.1 Measuring the Usage of Cross-Class (Robust) Features

Consider a $K$-class classification task where data from each class $y \sim \mathcal{Y}$ has a data distribution $x \sim \mathcal{D}_y$. Let $f(\cdot) = Wg(\cdot)$ represent a classifier, where $g$ is the feature extractor with $n$ dimension and $W \in \mathbb{R}^{K \times n}$ is the linear layer. For a given a sample $x$ from the $i$-th class, the output logit for the $i$-th class is $f(x)_i = W[i]^T g(x) = \sum_{j=1}^{n} g(x)_j W[i,j]$, where $W[i]$ is the $i$-th row of $W$. Intuitively, $g(x)_j W[i,j]$ represents how the $j$-th feature influences the logit of the $i$-th class prediction of $f(x)$. Thus we use $A_i(x) = (g(x)_1 W[i,1], \cdots, g(x)_n W[i,n])$ as the *attribution vector* for the sample $x$ on class $i$, where the $j$-th element denotes the weight of the $j$-th feature.

**Characterizing Cross-class Features** We consider the similarity of attribution vectors. If the attribution vectors of samples $x_1$ and $x_2$ are highly similar, the model tends to use more features shared by them when calculating their logits on their classes. On the other hand, if the attribution vectors of $x_1$ and $x_2$ are almost orthogonal, the model uses fewer shared features or they just do not share features.

This observation can be generalized to the classes. We model the feature attribution vector of a given class as the average of the vectors of the test samples in this class. Further, since we are considering feature attribution in the context of adversarial robustness, we only consider the attribution of **robust features** (Tsipras et al., 2018) for classifying adversarial examples. Thus, we craft adversarial examples and analyze their attributions to measure the usage of shared robust features.

As discussed, we can measure the usage of cross-class robust features shared by two given classes with the similarity of their attribution vectors. Therefore, we construct the *feature attribution correlation matrix* using the cosine similarity between the attribution vectors: $C[i,j] = \frac{A^i \cdot A^j}{\|A^i\|_2 \cdot \|A^j\|_2}$.

The complete algorithm of calculating matrix $C$ is shown in Algorithm 2 in Appendix. For two classes indexed by $i$ and $j$, $C[i,j]$ denotes the similarity of their feature attribution vector, which a higher value indicates the model uses more features shared by these classes.

**Numerical Metric** To further support our claims, we propose a numerical metric named **C**lass **A**ttribution **S**imilarity (CAS) defined on the correlation matrix $C$: $CAS(C) = \sum_{i \neq j} \max(C[i,j], 0)$.

The max function is used since we only focus on the positive correlations, and the negative elements are small (see Figure 2) and do not affect our analysis. CAS can quantitatively reflect the usage of cross-class features for a certain checkpoint.

## 3.2 Characterizing Robust Overfitting through Cross-Class Features

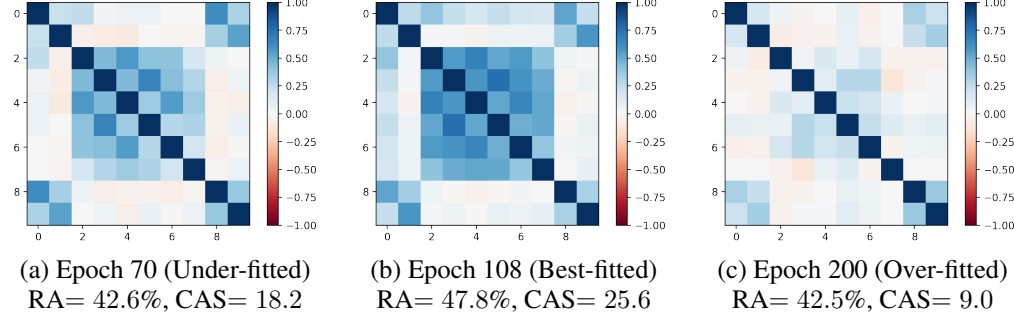

(a) Epoch 70 (Under-fitted)
RA= 42.6%, CAS= 18.2

(b) Epoch 108 (Best-fitted)
RA= 47.8%, CAS= 25.6

(c) Epoch 200 (Over-fitted)
RA= 42.5%, CAS= 9.0

Figure 2: Feature Attribution Correlation Matrix of models at different stages in AT, with their robust accuracy (RA) and CAS.

Based on the proposed measurement, we first visualize the feature attribution correlation matrices of vanilla AT. The model is trained on the CIFAR-10 dataset (Krizhevsky et al., 2009) using PreActResNet-18 (He et al., 2016) for 200 epochs, and it achieved its best test robust accuracy at the 108th epoch. More details can be found in Section 5.2. As shown in Figure 2, the model demonstrates a fair overlapping effect on feature attribution at the 70th epoch. Specifically, there

are several non-diagonal elements $C[i, j]$ in the correlation matrix $C$ that exhibit a relatively large value (in deeper blue), which indicates that the model leverages more features shared by the classes indexed by $i$ and $j$ when classifying adversarial examples from these two classes. Therefore, the model has already learned several cross-class features in the initial stage of AT. Moreover, when the model achieves its best robustness at the 108th epoch, the overlapping effect on feature attribution becomes clearer, with more non-diagonal elements in $C$ exhibiting larger values. This is also verified by the increase in CAS. However, at the end of AT, where the model is overfitted, the overlapping effect significantly decays, which indicates the model uses fewer cross-class features. We provide more correlation matrices of the model at different epochs in Appendix B.

This surprising effect motivates us to propose the following interpretation of robust overfitting. We identify two kinds of learning mechanisms in AT: (1) Learning *class-specific* features, *i.e.*, the features that are exclusive to only one class; (2) Learning *cross-class features*, *i.e.*, the same or similar features shared by more than one class. During the initial phase of AT, the model simultaneously learns exclusive class-wise features and cross-class features. Both of these features help achieve robust generalization and reduce training robust loss. However, once the training robust error is reduced to a certain degree, it becomes difficult for the model to further decrease it by optimizing cross-class features. This is because the features shared with other classes tend to raise positive logit on the shared classes. Thus, to further reduce the training robust loss, the model begins to reduce its reliance on cross-class features and bias more weight on class-specific features. Meanwhile, due to the strong memorization ability of DNNs in AT (Dong et al., 2022), the model also memorizes the training samples along with their corresponding adversarial examples, which further reduces the training robust error. This overall procedure can optimize training robust error but can also hurt test robust error by forgetting cross-class features, leading to a decrease in test robust accuracy and resulting in robust overfitting. We further provide more comprehensive empirical evidence on this explanation in the following.

### 3.3 MORE COMPREHENSIVE STUDY

In this section, we conduct a more comprehensive study of our proposed understanding with various empirical evidence.

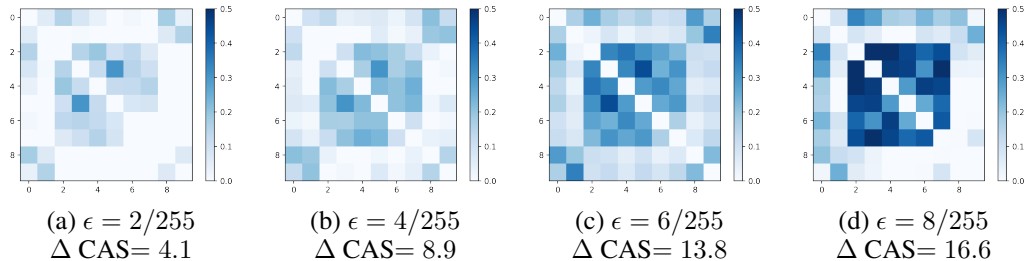

(a) $\epsilon = 2/255$     (b) $\epsilon = 4/255$     (c) $\epsilon = 6/255$     (d) $\epsilon = 8/255$
$\Delta$ CAS$= 4.1$     $\Delta$ CAS$= 8.9$     $\Delta$ CAS$= 13.8$     $\Delta$ CAS$= 16.6$

Figure 3: The **differences** between the feature attribution correlation matrices ($C_{\text{best}} - C_{\text{last}}$) and CAS of the best and the last checkpoint with various training perturbation bound $\epsilon$.

**Comparing with different perturbation bound** $\epsilon$ In Figure 3, we show the differences of the feature attribution correlation matrices and CAS between the best and last checkpoint of AT with various perturbation bounds $\epsilon$. The difference between the two matrices indicates how many cross-class features are abandoned by the model from the best checkpoint to the last. When $\epsilon = 2/255$, there is no significant difference between the best and last checkpoint. This is consistent with the fact that AT with small $\epsilon$ does not severely overfit, as shown in Figure 1. However, as $\epsilon$ increases, AT exhibits more overfitting effects, and the difference becomes more significant. This also verifies that the forgetting of cross-class features is a key factor of robust overfitting.

We offer a further explanation as to why larger perturbations cause more severe robust overfitting. Intuitively, AT with a larger perturbation bound $\epsilon$ results in a more rigid robust loss. During AT with a large $\epsilon$, cross-class features are more likely to be eliminated by the model to reduce training robust loss. We prove this claim in Theorem 1 in the next section.

While we mainly focus on AT with practically used $\epsilon$ (*e.g.*, $[0, 8/255]$ for $\ell_\infty$-AT), it is also observed that for extremely large $\epsilon(> 8/255)$, the effect of robust overfitting begins to decline (Wei et al., 2023a). Our interpretation is also compatible with this phenomenon, which we discuss in Appendix C. In brief, cross-class features are more sensitive under extremely large $\epsilon$, making them even harder to learn at the initial stage resulting in fewer forgetting of these features in the latter stage of AT.

**Comparing on other norms, datasets, and architectures** We also investigate this effect in AT with $\ell_2$-norm, CIFAR-100 (Krizhevsky et al., 2009) and TinyImagenet datasets (mnmoustafa, 2017), and vision transformer architecture (Touvron et al., 2021). Due to the space limitation, we leave the compared feature attribution correlation matrices and their corresponding CAS in Appendix D. Interestingly, similar to the effect demonstrated in $\ell_\infty$-norm AT with convolutional architecture (PreActResNet-18) on the CIFAR-10 dataset, in these settings the best checkpoints consistently use more cross-class features than the last checkpoints, verifies that our proposed understanding also holds in AT under various settings.

**Visualization of saliency map** To further analyze the feature attribution of AT at different stages, we compare the saliency maps on several examples that are correctly classified by the best but misclassified by the last checkpoint under adversarial attack, as shown in Figure 4 (a). The saliency map is derived by Grad-CAM (Selvaraju et al., 2017) on the true labeled classes. Taking the first column as an example, the classes *automobile* and *truck* share similar features like *wheels*. The best checkpoint pays more attention to the overall car including the wheel, whereas the last checkpoint solely focuses on the circular car roof that is exclusive to automobiles. This explains why the last checkpoint misclassifies this sample, for it only identifies this local feature for the true class and does not leverage holistic feature information from the image. The other five samples also exhibit a similar effect, with exclusive features being the mane for *horse*, the frog eyes for *frog*, the feather for *bird*, and the antlers for *deer*. Since the final checkpoint makes decisions based only on these limited features, it fails to leverage comprehensive features for classification, making the model more vulnerable to adversarial attacks.

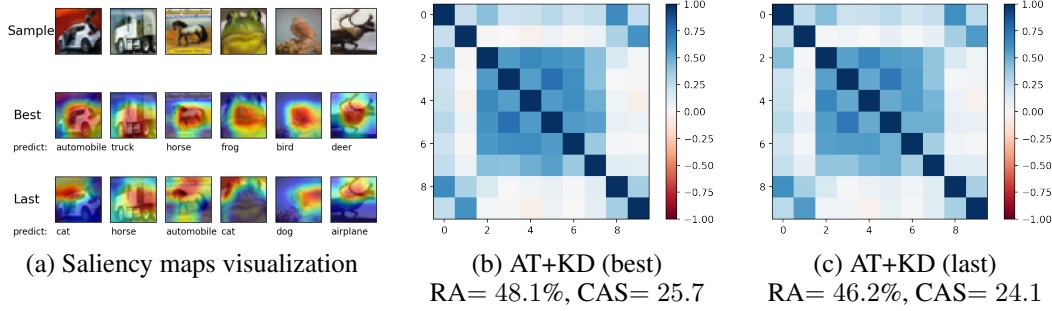

(a) Saliency maps visualization

(b) AT+KD (best)
RA= 48.1%, CAS= 25.7

(c) AT+KD (last)
RA= 46.2%, CAS= 24.1

Figure 4: (a): Visualization of saliency map with GradCAM. The top row shows the original sample, and the middle and bottom rows show the saliency map on adversarial examples of the best and the last checkpoint, respectively. (b), (c): the best and the last checkpoint of AT with knowledge distillation, and their Robust Accuracy (RA) and CAS.

**Knowledge distillation mitigates robust overfitting** Our understanding can also explain why knowledge distillation is a helpful technique for mitigating robust overfitting. In the process of AT with knowledge distillation, the teacher model adeptly captures the cross-class features present in the training data, and provides more precise labels by considering both class-specific and cross-class features. This stands in contrast to vanilla AT with one-hot labels, which primarily emphasizes class-specific features and may inadvertently suppress cross-class features in the model weights. The incorporation of cross-class features, backed by both our empirical findings and theoretical insights highlighting their significance for enhanced robustness, enables knowledge distillation to effectively mitigate robust overfitting by preserving these crucial features. We present a comparison between the best and last checkpoint of AT with knowledge distillation in Figure 4 (b) and (c), where no significant differences between the two matrices, nor a large gap between their CAS. Therefore, we conclude that AT with knowledge distillation helps mitigate robust overfitting by identifying cross-class features and providing more precise labels by considering these features.

# 4 THEORETICAL INSIGHTS

In this section, we provide theoretical evidence with a synthetic data model.

## 4.1 DATA DISTRIBUTION AND HYPOTHESIS SPACE

In this theoretical framework, we introduce a data distribution with class-specific and cross-class feature decomposition, along with a hypothesis space with linear functions.

**Data distribution** We consider a tertiary classification task, where each class owns an exclusive feature $x_{E,i}$, and every two classes have a shared cross-class feature $x_{C,j}$. The features for each sample can be formulated as $\{x_{E,j}, x_{C,j} | 1 \le j \le 3\} \in \mathbb{R}^6$. The data distribution is similar to the model applied in robust and non-robust features (Tsipras et al., 2018), but we only focus on the inner relation between robust features (class-specific or cross-class) and omit the non-robust features. As discussed above, we model the data distribution of the $i$-th class $y_i$ as $\mathcal{D}_i =$:

$$x_{E_j} \mid y = i \sim \begin{cases} \mathcal{N}(\mu, \sigma^2) & \text{if } j = i \\ 0 \text{ w.p. } 1 & \text{if } j \neq i \end{cases}, \quad x_{C_j} \mid y = i \sim \begin{cases} \mathcal{N}(\mu, \sigma^2) & \text{if } j \neq i \\ 0 \text{ w.p. } 1 & \text{if } j = i \end{cases}, \quad (5)$$

where $i \in \{1, 2, 3\}$, and $\mu, \sigma > 0$. We also assume $\sigma < \sqrt{\pi}\mu$ to control the variance.

**Hypothesis space** We introduce a linear model $f(x)$ in this classification task, which gives $i$-th logit for sample $x$ by $f(x)_i = \sum_j w_{i,j}^E x_{E,j} + \sum_j w_{i,j}^C x_{C,j}$. However, there are 6 parameters in the data samples, making this linear model hard to analyze. Thus we simplify the model based on the following observations. First, we can simply keep $w_{i,j}^E = 0$ for $i \neq j$ and $w_{i,i}^C = 0$ due to the corresponding data distribution is identity to 0. Further, we set $w_{1,1}^E = w_{2,2}^E = w_{3,3}^E = w_1$ and $w_{i,j}^C = w_2 (i \neq j)$ due to symmetry. Finally, we assume $w_1, w_2 \ge 0$ since $\mu > 0$. Overall, the hypothesis space is $\{f_{\boldsymbol{w}} : \boldsymbol{w} = (w_1, w_2), w_1, w_2 \ge 0\}$ and $f_{\boldsymbol{w}}(x)$ calculates its $i$-th logit by

$$f_{\boldsymbol{w}}(\boldsymbol{x})_i = w_1 x_{E,i} + w_2 (x_{C,j_1} + x_{C,j_2}), \quad \text{where} \quad \{j_1, j_2\} = \{1, 2, 3\} \backslash \{i\}. \quad (6)$$

Now we consider adversarially training $f_{\boldsymbol{w}}$ with $\ell_\infty$-norm perturbation bound $\epsilon < \frac{\mu}{2}$. We also add a regularization term $\frac{\lambda}{2}\|\boldsymbol{w}\|_2^2$ to the overall loss function, which can be modeled as $\mathbb{E}_{i \sim p_y}\{\mathbb{E}_{x \sim \mathcal{D}_i} \max_{\|\delta\|_p \le \epsilon} \ell(w; x + \delta)\} + \frac{\lambda}{2}\|\boldsymbol{w}\|_2^2$, where

$$\ell(w; x + \delta) = \max_{\|\delta\|_\infty \le \epsilon} (\max_{j \neq i} f_{\boldsymbol{w}}(x + \delta)_j - f_{\boldsymbol{w}}(x + \delta)_i). \quad (7)$$

## 4.2 MAIN RESULTS

**Cross-class features are more sensitive to robust loss** We show that under the robust training loss (7), the model tends to abandon $x_C$ by setting $w_2 = 0$ if $\epsilon$ is larger than a certain threshold. However, any $\epsilon \in (0, \frac{\mu}{2})$ returns a positive $w_1$, as stated in Theorem 1. This result indicates that cross-class features are more sensitive to robust loss and are more likely to be eliminated in AT compared to class-specific features, even when they share the same mean value $\mu$.

**Theorem 1** *There exists a $\epsilon_0 \in (0, \frac{1}{2}\mu)$, for AT by optimizing the robust loss (7) with $\epsilon \in (0, \epsilon_0)$, the output function obtains $w_2 > 0$; for AT with $\epsilon \in (\epsilon_0, \frac{1}{2}\mu)$, the output function returns $w_2 = 0$. By contrast, AT with $\epsilon \in (0, \frac{1}{2}\mu)$ always obtains $w_1 > 0$.*

This claim is also consistent with our discussion on AT with different $\epsilon$ in Section 3.3. Recall that AT with larger $\epsilon$ tends to compress more cross-class features as shown in Figure 3. This observation can be verified by Theorem 1 that cross-class features are more likely to be eliminated during AT with larger $\epsilon$, which causes more severe robust overfitting.

**Cross-class features are helpful for robust classification** Although decreasing the value of $w_2$ may reduce the robust training error, we demonstrate in Theorem 2 that using a positive $w_2$ is always more beneficial for robust classification than simply setting $w_2$ to 0.

**Theorem 2** *For any class $y$, consider weights $w_1 > 0$, $w_2 \in [0, w_1]$, and $\epsilon \in (0, \frac{\mu}{2})$. When sampling $x$ from the distribution of class $y$, increasing the value of $w_2$ enhances the possibility of the model assigning a higher logit to class $y$ than to any other classes $y' \neq y$ under adversarial attack. In other words, the probability $\Pr_{x \sim \mathcal{D}_y}[f_w(x + \delta))_y > f_w(x + \delta)_{y'}, \forall \delta : \|\delta\|_\infty \leq \epsilon]$ monotonically increases with $w_2$ within the range $[0, w_1]$.*

**Knowledge distillation preserves cross-class features** Finally, we show that knowledge distillation helps preserve the cross-class features, which provide a justification on why this method can alleviate robust overfitting. Note that due to the symmetry of distributions and weights among classes, we apply label smoothing to simulate knowledge distillation (which we justify in Section E.4 in detail) and rewrite the robust loss as $\mathbb{E}_{i \sim p_y}\{\mathbb{E}_{x \sim \mathcal{D}_i} \max_{\|\delta\|_p \leq \epsilon} \ell_{\mathrm{LS}}(w; x + \delta)\} + \frac{\lambda}{2}\|\boldsymbol{w}\|_2^2$, where

$$\ell_{\mathrm{LS}}(w; x + \delta) = (1 - \beta)[\max_{\|\delta\|_\infty \leq \epsilon} (\max_{j \neq i} f_{\boldsymbol{w}}(x + \delta)_j - f_{\boldsymbol{w}}(x + \delta)_i)] - \frac{\beta}{2} \sum_{j \neq i} f_{\boldsymbol{w}}(x + \delta)_j \quad (8)$$

and $\beta < \frac{1}{3}$ is the interpolation ratio of label smoothing. In Theorem 3 and Corollary 1, we show that not only the label smoothed loss (8) enables larger perturbation bound $\epsilon$ for utilizing cross-class features, but also returns larger $w_2$. This explains that preserving the cross-class features is the reason why knowledge distillation helps mitigate robust overfitting.

**Theorem 3** *Consider AT with knowledge distillation loss (8). There exists an $\epsilon_1 \in (0, \frac{\mu}{2})$ with $\epsilon_1 > \epsilon_0$ derived in Theorem 1, such that for $\epsilon \in (0, \epsilon_1)$, the output function obtains $w_2 > 0$; for $\epsilon \in (\epsilon_1, \frac{1}{2}\mu)$, the output function returns $w_2 = 0$.*

**Corollary 1** *Let $w_2^*(\epsilon)$ be the value of $w_2$ returned by AT with (7), and $w_2^{LS}(\epsilon)$ be the value of $w_2$ returned by label smoothed loss (8). Then, for $\epsilon \in (0, \epsilon_1)$, we have $w_2^{LS}(\epsilon) > w_2^*(\epsilon)$.*

All proofs can be found in Appendix E. To summarize, our theoretical analysis demonstrates that cross-class features are more sensitive to robust loss, yet helpful for robust classification. We also show that knowledge distillation can mitigate robust overfitting by preserving the cross-class features.

## 5 BETTER KNOWLEDGE DISTILLATION FURTHER IMPROVES ROBUSTNESS

In this section, we propose an improved knowledge distillation approach to further enhance adversarial robustness and mitigate robust overfitting in AT.

### 5.1 WEIGHT AVERAGE GUIDED KNOWLEDGE DISTILLATION

Based on our understanding which explains that knowledge distillation can help alleviate robust overfitting by preserving cross-class features, we aim to introduce a better teacher model for knowledge distillation which can characterize more precise cross-class feature distribution. Motivated by the fact that weight-averaged models exhibit better robustness in AT (Wang & Wang, 2022), we propose leveraging the weight-averaged model as the teacher model for knowledge distillation, which also outperforms vanilla knowledge distillation in terms of computational cost since it does not require a pre-trained robust model.

The loss function is similar to Equation (3), but with the robust-trained teacher replaced by an averaged model and the standard-trained teacher removed. The loss function can be formulated as

$$\max_{\|\delta\|_p \leq \epsilon} \tilde{\ell}(\theta \,; \bar{\theta}, x + \delta, y), \quad \text{where } \tilde{\ell}(\theta \,; \bar{\theta}, x + \delta, y) = (1 - \lambda)\ell_{CE}(f(\theta, x + \delta), y) + \lambda \mathcal{KD}(f(\theta, x + \delta), f(\bar{\theta}, x + \delta))$$

$$(9)$$

where $\bar{\boldsymbol{\theta}}$ represents the parameter of the weight averaged model in AT, and $\lambda$ is the interpolation ratio.

However, some modifications are needed. First, the weight-averaged model requires warm-up before it is applied for knowledge distillation. Second, if we directly start to apply loss (9) at a specific checkpoint, we observe a catastrophic forgetting that the test accuracy drops significantly, which may be due to a drastic change in the loss function. Therefore, we introduce a (piecewise) linear

Table 1: Comparison of our method with vanilla AT and AT+KDSWA.

| Dataset | Method | Robust Acc. (%) | | Clean Acc. (%) | |
|---------|--------|------|------|------|------|
| | | Best | Last | Best | Last |
| CIFAR-10 | AT | 47.8 ±0.2 | 42.5 ±0.2 | 82.7 ±0.5 | 84.5 ±0.3 |
| | AT + KDSWA | 49.8 ±0.4 | 49.6 ±0.2 | 83.8 ±0.6 | 84.7 ±0.4 |
| | AT + WAKE | **50.4** ±0.3 | **50.1** ±0.2 | **83.9** ±0.3 | **84.9** ±0.3 |
| CIFAR-100 | AT | 24.7 ±0.2 | 19.6 ±0.3 | 55.6 ±0.5 | 57.4 ±0.2 |
| | AT + KDSWA | 26.1 ±0.3 | 25.7 ±0.2 | 58.6 ±0.5 | 59.1 ±0.2 |
| | AT + WAKE | **26.8** ±0.3 | **26.5** ±0.2 | **59.5** ±0.4 | **59.7** ±0.1 |
| Tiny-Imagenet | AT | 18.0 ±0.3 | 14.4 ±0.4 | 45.5 ±0.6 | 48.3 ±0.4 |
| | AT + KDSWA | 19.9 ±0.3 | 19.4 ±0.3 | 49.7 ±0.4 | 50.4 ±0.3 |
| | AT + WAKE | **20.4** ±0.2 | **19.9** ±0.2 | **50.2** ±0.3 | **50.8** ±0.2 |

scheduler to set the $\lambda$ in (9) to stabilize the training process. The $\lambda$ is set to 0 initially, and then gradually increases to the target after a certain checkpoint. Overall, we name our proposed method as **W**eight **A**verage guided **K**nowledg**E** distillation (WAKE), and the complete algorithm is elaborated in Algorithm 2 in Appendix G.

## 5.2 EXPERIMENT

**Settings** We conduct experiment on CIFAR-{10, 100} (Krizhevsky et al., 2009) and Tiny-Imagenet (mnmoustafa, 2017) datasets using PreActResNet-18 (PRN-18) (He et al., 2016) model. Following the best settings in (Rice et al., 2020), we train the model using SGD with a momentum of 0.9, weight decay of $5 \times 10^{-4}$, and an initial learning rate of 0.1. We compare our method with vanilla AT and KDSWA (Chen et al., 2021). Following the same settings as in AT+KDSWA, we train 200 epochs for CIFAR datasets and 100 epochs for Tiny-Imagenet. During AT, we apply a 10-step PGD attack with a $\ell_\infty$-norm perturbation bound $\epsilon = 8/255$ and a step size of $\alpha = 2/255$. For WAKE, we set the maximum interpolation ratio to $\lambda = 0.8$ and the knowledge distillation temperature to $T = 2$ for WAKE. The distillation warm-up starts and ends at epochs 90 and 110 for CIFAR datasets and at epochs 40 and 60 for Tiny-Imagenet. We use AutoAttack (AA.) (Croce & Hein, 2020) for reliable robustness evaluation, and conduct five independent experiments for each method and report the mean result and standard deviation. We also conduct experiments on AT with $\ell_2$-norm, CIFAR-100 and TinyImagenet dataset, and vision transformer architecture, and show the results in Appendix F.

**Improving robustness and alleviating overfitting** Table 1 shows the overall comparison of our method and the baselines. From the results, we can see that in terms of adversarial robustness, our AT+WAKE outperforms vanilla AT and AT+KDSWA in all three datasets, both at the best and last checkpoints, since the better teacher models can characterize more precise cross-class features. In addition, regarding clean accuracy, AT+WAKE also outperforms vanilla AT and AT+KDSWA on these datasets, showing that our method achieves a better clean *vs.* robustness trade-off (Tsipras et al., 2018; Zhang et al., 2019). Overall, our proposed WAKE further improves adversarial robustness and mitigates robust overfitting in AT, and also has the advantage of lower computational cost.

## 6 CONCLUSION

In this paper, we provide a novel interpretation of robust overfitting in AT through the lens of feature attribution. We point out that during AT, in order to achieve lower training robust loss, the model's tendency to reduce its reliance on cross-class features is a key factor in robust overfitting. We empirically verify this claim by measuring the dependence on cross-class features of the model at different stages in AT under various settings, along with other empirical evidence including analysis of saliency maps and knowledge distillation-based methods. We also provide theoretical insights demonstrating that cross-class features are more sensitive to training robust loss, but are actually helpful for robust classification. Based on this understanding, we finally propose a weight-average guided knowledge distillation method that further boosts adversarial robustness.

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

## A ALGORITHM FOR CALCULATING THE FEATURE ATTRIBUTION CORRELATION MATRIX

We present the complete algorithm of calculating the feature attribution correlation matrix in Algorithm 2. For each class, we first calculate the feature attribution vectors for each test adversarial sample, then calculate the mean of these vectors as the feature attribution vector of this class. Finally, we calculate the cosine similarity of the vectors as the measure of cross-class feature usage for each pair of two classes.

---

**Algorithm 1:** Feature Attribution Correlation Matrix

**Input:** A DNN classifier $f$ with feature extractor $g$ and linear layer $W$; **Test** dataset
$\qquad D = \{D_y : y \in \mathcal{Y}\}$; Perturbation margin $\epsilon$;
**Output:** A correlation matrix $C$ measuring the cross-class feature usage
```
/* Record robust feature attribution                                */
```
**for** $y \in \mathcal{Y}$ **do**
$\quad A^y \leftarrow (0, \cdots, 0)$ `/* initialization as a n-dim vector            */`
$\quad$ **for** $x \in D_y$ **do**
$\qquad \delta \leftarrow \arg\max_{\|\delta\| \le \epsilon} \ell_{\text{CE}}(f(x+\delta), y)$ `/* untargeted PGD Attack        */`
$\qquad A^y \mathrel{+}= g(x+\delta) \odot W[y]$ `/* point-wise multiplication             */`
$\quad A^y \leftarrow A^y / |D_y|$ `/* Average                                       */`
**for** $1 \le i, j \le |\mathcal{Y}|$ **do**
$\quad C[i,j] \leftarrow \frac{A^i \cdot A^j}{\|A^i\|_2 \cdot \|A^j\|_2}$ `/* Cosine similarity                          */`
**return** $C$

---

## B MORE FEATURE ATTRIBUTION CORRELATION MATRICES AT DIFFERENT EPOCHS

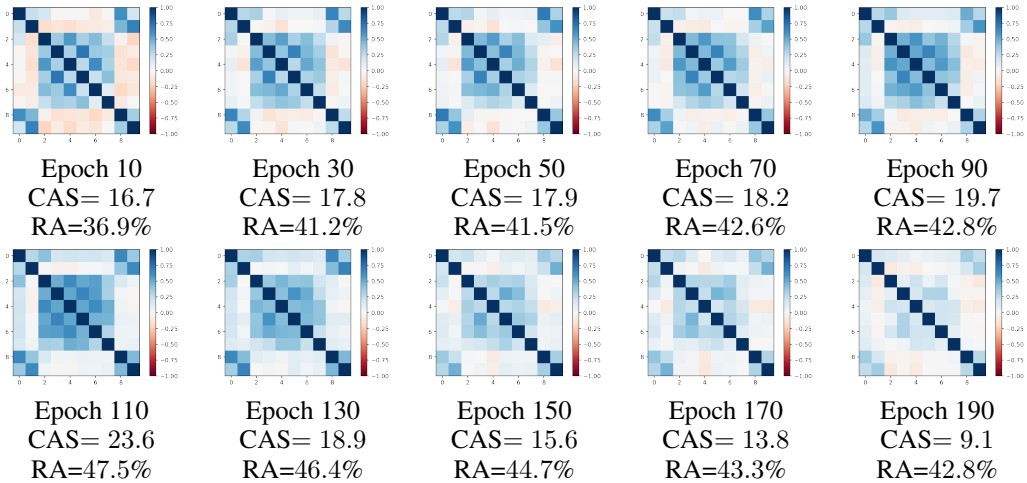

| Epoch 10 | Epoch 30 | Epoch 50 | Epoch 70 | Epoch 90 |
|---|---|---|---|---|
| CAS= 16.7 | CAS= 17.8 | CAS= 17.9 | CAS= 18.2 | CAS= 19.7 |
| RA=36.9% | RA=41.2% | RA=41.5% | RA=42.6% | RA=42.8% |

| Epoch 110 | Epoch 130 | Epoch 150 | Epoch 170 | Epoch 190 |
|---|---|---|---|---|
| CAS= 23.6 | CAS= 18.9 | CAS= 15.6 | CAS= 13.8 | CAS= 9.1 |
| RA=47.5% | RA=46.4% | RA=44.7% | RA=43.3% | RA=42.8% |

Figure 5: Feature attribution correlation matrices, and their corresponding robust accuracy (RA), CAS at different epochs.

We present more feature attribution correlation matrices at different epochs in Figure 5. The training detail is the same as that of our experiment (Section 5.2), and the test robust accuracy is plotted in Figure 1(b) (red line, $\epsilon = 8/255$). From the matrices we can see that at the initial stage of AT (10th - 90th Epochs), the model has already learned several cross-class features, and the overlapping effect of class-wise feature attribution achieves the highest at the 110th epoch among the shown matrices. However, for the later stages, where the model starts overfitting, this overlapping effect gradually vanishes, and the model tends to make decisions with fewer cross-class features.

## C    REGARDING EXTREMELY LARGE $\epsilon$

While our interpretation is consistent with the fact that for practically used $\epsilon \in [0, 8/255]$, larger $\epsilon$ leads to more significant robust overfitting in AT, it is also compatible with the phenomenon of for extremely large $\epsilon (> 8/255)$, the effect of robust overfitting begins to decline (Wei et al., 2023a). We justify this below.

Recall that our main interpretation for robust overfitting is that during the initial stage of AT, the model learns both class-specific and cross-class features. As training progresses and the robust loss decreases, the model begins to forget cross-class features, which leads to robust overfitting. Regarding AT with extremely large $\epsilon$ , as we proved in Theorem 1, the more rigid robust loss makes the model even harder to learn cross-class features at the initial stage of AT. Given that fewer cross-class features are learned, the forgetting effect of these features is weakened, thus mitigating robust overfitting.

This claim is verified by the following study. We conduct additional experiments on AT with extremely large perturbation bounds $\epsilon = 12/255$ and $16/255$, and compare them with $\epsilon = 8/255$. We report the CAS and robust accuracy at the **10th, best, and last epochs** in the following table.

Table 2: Comparison of robust accuracy (RA) and CAS on AT with large $\epsilon$.

| Epoch | 10 | Best | Last |
|---|---|---|---|
| $\epsilon$ for AT | CAS / RA | CAS / RA | CAS / RA |
| 8/255 | 16.7/36.9% | 25.6/47.8% | 9.0/42.5% |
| 12/255 | 15.6/29.8% | 18.9/38.7% | 8.7/34.1% |
| 16/255 | 14.4/23.8% | 17.5/31.3% | 8.4/28.1% |

The table shows that the CAS (usage of cross-class features) of large $\epsilon$ is less than that of $\epsilon = 8/255$ during the initial stage of AT (10th Epoch). This verifies our claim that the more rigid robust loss of large $\epsilon$ makes it even harder for the model to learn cross-class features at the initial stage of AT. Furthermore, the CAS of the best Epoch for large $\epsilon$ is significantly smaller than that of $\epsilon = 8/255$, further supporting our claim that these models struggle to learn cross-class features. Comparing the gap of CAS between the best and last epochs, we find that the gap for large $\epsilon$ is smaller than that of $\epsilon = 8/255$, which is consistent with the gap between the best and last robust accuracy. Therefore, we can conclude that the mitigation of robust overfitting with large $\epsilon$ can be explained by the less forgetting of cross-class features, which is compatible with our interpretation.

## D    MORE COMPARISON UNDER VARIOUS SETTINGS

### D.1    COMPARISON ON MORE DATASETS

We illustrate the comparison of the feature attribution correlation matrices and the corresponding robust accuracy and CAS of the best checkpoint and the last checkpoint on the **CIFAR-100** and the **TinyImagenet** datasets in Figure 6 and Figure 7, respectively. We can see that there are still significant differences between matrices and CAS derived from the best and the last checkpoint of AT on other datasets.

### D.2    COMPARISON ON $\ell_2$-NORM AT

We show the comparison of the feature attribution correlation matrices of the best checkpoint and the last checkpoint of $\boldsymbol{\ell_2}$**-norm AT** ($\epsilon = 128/255$) on CIFAR-10 dataset in Figure 8 (a)(b). We can see that there are still significant differences between matrices and CAS derived from the best and the last checkpoint of $\ell_2$-norm AT.

### D.3    COMPARISON ON TRANSFORMER ARCHITECTURE

We show the comparison of the feature attribution correlation matrices of the best checkpoint and the last checkpoint of AT on CIFAR-10 dataset with **Vision Transformer architecture** (Deit-Ti Touvron

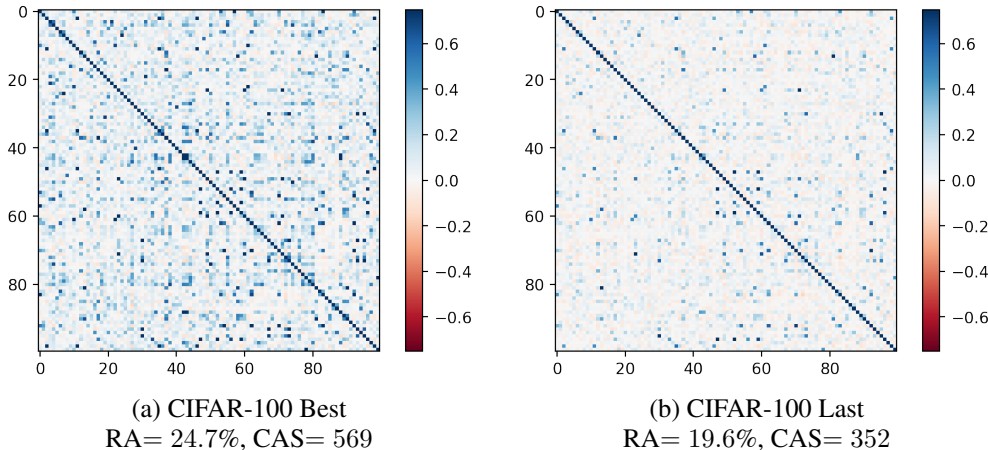

(a) CIFAR-100 Best
RA= 24.7%, CAS= 569

(b) CIFAR-100 Last
RA= 19.6%, CAS= 352

Figure 6: Feature attribution correlation matrices on CIFAR-100 dataset.

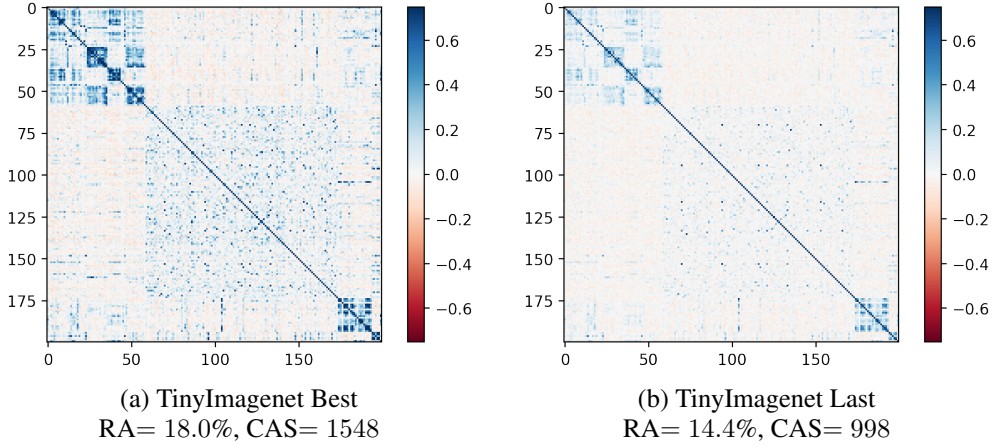

(a) TinyImagenet Best
RA= 18.0%, CAS= 1548

(b) TinyImagenet Last
RA= 14.4%, CAS= 998

Figure 7: Feature attribution correlation matrices on $\ell_2$-norm AT and Visual Transformer architecture.

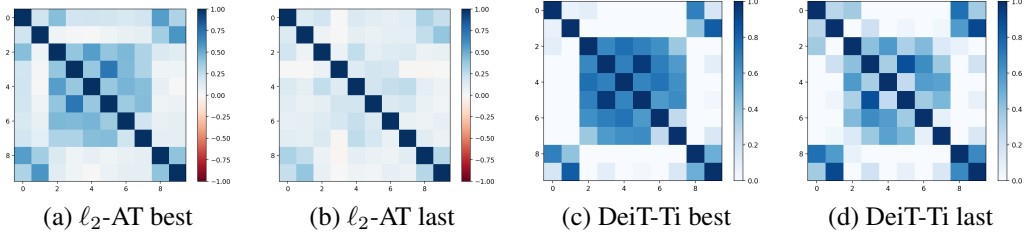

(a) $\ell_2$-AT best     (b) $\ell_2$-AT last     (c) DeiT-Ti best     (d) DeiT-Ti last

Figure 8: Feature attribution correlation matrices on $\ell_2$-norm AT and Visual Transformer architecture.

et al. (2021)) in Figure 8 (c)(d). We can see that there are still significant differences between matrices and CAS derived from the best and the last checkpoint of AT with transformer architecture.

## D.4 INSTANCE-WISE ANALYSIS

We also conduct a similar study by calculating the feature attribution correlation matrices for the best and the last checkpoints of $\ell_\infty$ and $\ell_2$-AT and their corresponding CAS **instance-wisely**, and the results are shown in Figure 9. When considering classes $i$ and $j$, for each sample $x$ from class $i$, we identify its most similar counterpart $x'$ from class $j$. We then calculate their cosine similarity and average the results over all samples in class $i$.

In this context, $x'$ can be interpreted as the sample in class $j$ that shares the most cross-class features with $x$ among all samples in class $j$. This metric provides a meaningful way to quantify the utilization of cross-class features. We did attempt to average over all sample pairs $(x, x')$ in classes $i$ and $j$, but due to high variance among samples, each element in the correlation matrix $C$ hovered near 0 throughout all epochs in adversarial training, rendering it unable to provide meaningful information.

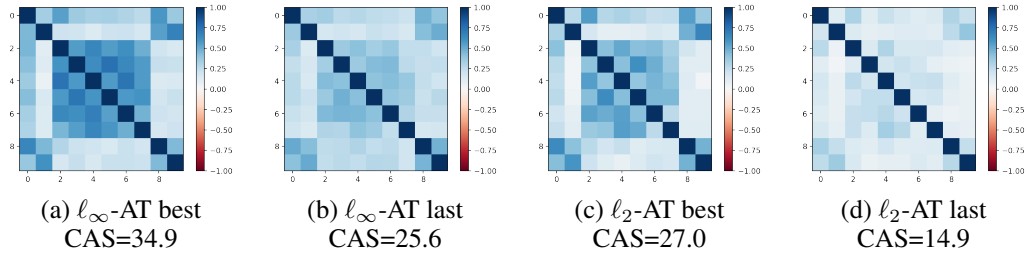

(a) $\ell_\infty$-AT best
CAS=34.9

(b) $\ell_\infty$-AT last
CAS=25.6

(c) $\ell_2$-AT best
CAS=27.0

(d) $\ell_2$-AT last
CAS=14.9

Figure 9: **Instance-wise** feature attribution correlation matrices

Consistent with the results for class-wise attribution vectors, it is still observed that there is a significant decrease in the usage of cross-class features from the best checkpoint to the last for both $\ell_\infty$ and $\ell_2$-AT. This observation further substantiates our understanding of robust overfitting.

## D.5 REGULAR TRAINING

We also extend our experimental scope to include regular training on the CIFAR-10 dataset. The experimental settings mirror those outlined in Section 5, with the sole distinction being the absence of perturbations in regular training. The results are shown in Figure 10. Specifically, considering that regular training prioritizes natural generalization and exhibits minimal robustness, we have calculated the feature attribution vectors using clean examples. These vectors were computed for epochs $\{50, 100, 150, 200\}$. Notably, the results reveal a lack of clear differences between them, particularly in the latter stages (150th and 200th), where the training tends to converge. This observation is consistent with the characteristic of regular training, which typically does not exhibit overfitting.

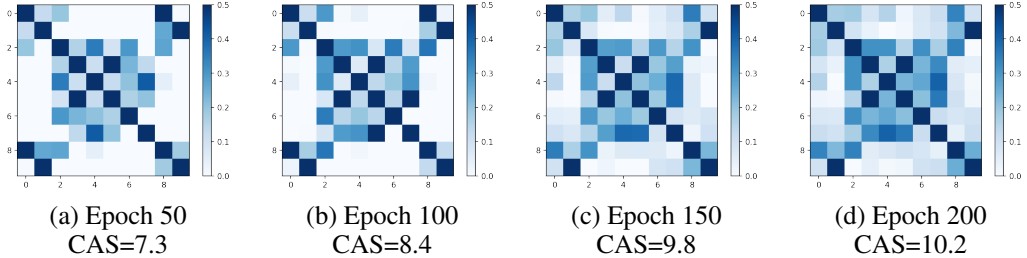

(a) Epoch 50
CAS=7.3

(b) Epoch 100
CAS=8.4

(c) Epoch 150
CAS=9.8

(d) Epoch 200
CAS=10.2

Figure 10: Feature attribution correlation matrices for **regular training** at different stages. Color bar scaled to $[0, 0.5]$.

### D.6 CATASTROPHIC OVERFITTING IN FAST-AT

In addition to robust overfitting in adversarial training, there is also a phenomenon called *Catastrophic Overfitting* (Wong et al., 2020) observed in Fast (single step) adversarial training, where the model quickly decreases its robustness after a certain epoch of training. We also extend our investigations to include Fast-AT for the CIFAR-10 dataset, employing an $\ell_\infty$-norm perturbation bound of $\epsilon = 8/255$. The feature attribution correlation matrices before and after the catastrophic overfitting are shown in Figure 11. It is clear that after catastrophic overfitting, there is a significant reduction in the usage of cross-class features. This observation aligns with our understanding, indicating that the model also tends to forget cross-class features after exhibiting catastrophic overfitting.

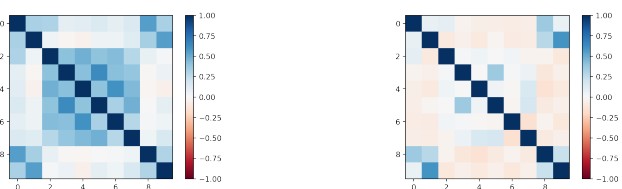

(a) Before Catastrophic Overfitting     (b) After Catastrophic Overfitting

Figure 11: Feature attribution correlation matrices for **Fast adversarial training** before and after catastrophic overfitting happens.

## E PROOFS FOR THEOREMS

### E.1 PRELIMINARIES

First we present some preliminaries, and then review the data distribution, the hypothesis space and the optimization objective.

**Notations** Let $\mathcal{N}(\mu, \sigma)$ be the normal distribution with mean $\mu$ and variance $\sigma^2$. We denote $\phi(x) = \frac{1}{\sqrt{2\pi}} e^{-\frac{x^2}{2}}$ and $\Phi(x) = \int_{-\infty}^{x} \frac{1}{\sqrt{2\pi}} e^{-\frac{t^2}{2}} \mathrm{d}t = \mathrm{Pr}.(\mathcal{N}(0,1) < x)$ as its probability density function and distribution function.

**Data distribution** For $i \in \{1, 2, 3\}$, the sample of the $i$-th class is

$$(x_{E,1}, x_{E,2}, x_{E,3}, x_{C,1}, x_{C,2}, x_{C,3}) \in \mathbb{R}^6, \tag{10}$$

follows a distribution

$$\begin{cases} x_{E,j}|(y_i = j) \sim \mathcal{N}(\mu, \sigma^2) \\ x_{E,j}|(y_i \neq j) = 0 \end{cases}, \quad \begin{cases} x_{C,j}|(y_i \neq j) \sim \mathcal{N}(\mu, \sigma^2) \\ x_{C,j}|(y_i = j) = 0 \end{cases}, \tag{11}$$

and $\mu, \sigma > 0$. We also assume $\sigma < \sqrt{\pi}\mu$ to control the variance.

**Hypothesis space** The hypothesis space is $\{f_{\boldsymbol{w}} : \boldsymbol{w} = (w_1, w_2), w_1, w_2 \geq 0\}$ and $f_{\boldsymbol{w}}(x)$ calculates its $i$-th logit by

$$f_{\boldsymbol{w}}(\boldsymbol{x})_i = w_1 x_{E,i} + w_2 (x_{C,j_1} + x_{C,j_2}), \quad \text{where} \quad \{j_1, j_2\} = \{1, 2, 3\}\backslash\{i\}. \tag{12}$$

**Optimization objective** Consider adversarially training $f_{\boldsymbol{w}}$ with $\ell_\infty$-norm perturbation bound $\epsilon < \frac{\mu}{2}$. We hope that given sample $x \sim \mathcal{D}_i$, under any perturbation $\{\delta : \|\delta\|_\infty \leq \epsilon\}$, the $f(x + \delta)_i$ is larger than any $f(x + \delta)_j$ as much as possible. We also add a regularization term $\frac{\lambda}{2}\|\boldsymbol{w}\|_2^2$ to the loss function.

Overall, the loss function can be formulated as

$$\mathcal{L}(f_{\boldsymbol{w}}) = \mathbb{E}_i[\mathbb{E}_{x \sim \mathcal{D}_i} \max_{\|\delta\|_\infty \leq \epsilon} (\max_{j \neq i} f_{\boldsymbol{w}}(x + \delta)_j - f_{\boldsymbol{w}}(x + \delta)_i)] + \frac{\lambda}{2}\|\boldsymbol{w}\|_2^2. \tag{13}$$

### E.2 Proof for Theorem 1

**Theorem 1** *There exists a $\epsilon_0 \in (0, \frac{1}{2}\mu)$, for AT by optimizing the robust loss (13) with $\epsilon \in (0, \epsilon_0)$, the output function obtains $w_2 > 0$; for AT with $\epsilon \in (\epsilon_0, \frac{1}{2}\mu)$, the output function returns $w_2 = 0$. By contrast, AT with $\epsilon \in (0, \frac{1}{2}\mu)$ always obtains $w_1 > 0$.*

To prove Theorem 1, we need the following lemmas.

**Lemma 1** *Suppose that $X, Y \sim \mathcal{N}(0, 1)$, and they are independent. Let $Z = \max\{X, Y\}$, then $\mathbb{E}[Z] = \frac{1}{\sqrt{\pi}}$.*

*proof.* Let $p(\cdot)$ and $F(\cdot)$ be the probability density function and distribution function of $Z$, respectively. Then, for any $z \in \mathbb{R}$,

$$F(z) = \Pr(Z < z) = \Pr(\max\{X, Y\} < z) = \Pr(X < z) \cdot \Pr(Y < z) = \Phi^2(z), \quad (14)$$

and we have

$$p(z) = F'(z) = [\Phi^2(z)]' = 2\phi(z)\Phi(z). \quad (15)$$

Thus,

$$
\begin{aligned}
\mathbb{E}[Z] &= \int_{-\infty}^{+\infty} 2z\phi(z)\Phi(z)dz \\
&= 2\int_{-\infty}^{+\infty} z \cdot \frac{1}{\sqrt{2\pi}}e^{-\frac{z^2}{2}}\left(\int_{-\infty}^{z} \frac{1}{\sqrt{2\pi}}e^{-\frac{t^2}{2}}dt\right)dz \\
&= -\frac{1}{\pi}\int_{-\infty}^{+\infty}\left(\int_{-\infty}^{z} e^{-\frac{t^2}{2}}dt\right)d(e^{-\frac{z^2}{2}}) \\
&= -\frac{1}{\pi}[e^{-\frac{z^2}{2}}\int_{-\infty}^{z} e^{-\frac{t^2}{2}}dt]_{-\infty}^{+\infty} + \frac{1}{\pi}\int_{-\infty}^{+\infty} e^{-\frac{z^2}{2}}e^{-\frac{z^2}{2}}dz \\
&= 0 + \frac{1}{\pi}\int_{-\infty}^{+\infty} e^{-z^2}dz = \frac{1}{\sqrt{\pi}}.
\end{aligned}
\quad (16)
$$

**Lemma 2** *Given $x = (x_{E,1}, x_{E,2}, x_{E,3}, x_{C,1}, x_{C,2}, x_{C,3}) \sim \mathcal{D}_1$, $\epsilon \in (0, \frac{\mu}{2})$ and $\mathbf{w} = (w_1, w_2)$, then $\delta = (-\epsilon, \epsilon, \epsilon, \epsilon, -\epsilon, -\epsilon)$ is a solution for $\delta = \arg \max_{\|\delta\|_\infty \leq \epsilon} [\max_{j \neq 1} f_{\mathbf{w}}(x + \delta)_j - f_{\mathbf{w}}(x + \delta)_1]$.*

*proof.* Denote $\delta = (\delta_{E,1}, \delta_{E,2}, \delta_{E,3}, \delta_{C,1}, \delta_{C,2}, \delta_{C,3})$. Note that for $x \sim \mathcal{D}_1$, we have $x_{E,2} = x_{E,3} = x_{C,1} = 0$. Then,

$$
\begin{aligned}
&\max_{j \neq 1} f_{\mathbf{w}}(x + \delta)_j - f_{\mathbf{w}}(x + \delta)_1 \\
&= \max_{j \in \{2,3\}} [w_1\delta_{E,2} + w_2\delta_{C,1} + w_2(x_{C,3} + \delta_{C,3}), \ w_1\delta_{E,3} + w_2\delta_{C,1} + w_2(x_{C,2} + \delta_{C,2})] \\
&\quad - w_1(x_{E,1} + \delta_{E,1}) - w_2(x_{C,2} + \delta_{C,2} + x_{C,3} + \delta_{C,3}) \\
&= w_2\delta_{C,1} + \max_{j \in \{2,3\}} [w_1\delta_{E,2} + w_2(x_{C,3} + \delta_{C,3}), \ w_1\delta_{E,3} + w_2(x_{C,2} + \delta_{C,2})] \\
&\quad - w_1(x_{E,1} + \delta_{E,1}) - w_2(x_{C,2} + \delta_{C,2} + x_{C,3} + \delta_{C,3}).
\end{aligned}
\quad (17)
$$

Since $w_1, w_2 \geq 0$, it is clear that $\delta_{E,1} = -\epsilon$, $\delta_{E,2} = \delta_{E,3} = \delta_{C,1} = \epsilon$ are the optimal values for maximizing (17). As for $\delta_{C,2}$ and $\delta_{C,3}$, to prove that $\delta_{C,2} = \delta_{C,2} = -\epsilon$ are the optimal values, by variable simplification ($a' = \delta_{C,2}, b' = \delta_{C,3}$) and dividing by $w_2$ we only need to show that

$$\max\{a + a', b + b'\} - a' - b' \leq \max\{a - \epsilon, b - \epsilon\} + 2\epsilon \quad (18)$$

under the constraint $|a'| \leq \epsilon$ and $|b'| \leq \epsilon$. Note that (18) is equivalent to

$$
\begin{aligned}
&\max\{a + a', b + b'\} - a' - b' \leq \max\{a, b\} + \epsilon \\
&\Leftrightarrow \max\{a + a', b + b'\} \leq \max\{a, b\} + a' + b' + \epsilon \\
&\Leftrightarrow \max\{a + a', b + b'\} \leq \max\{a + a' + b' + \epsilon, b + a' + b' + \epsilon\}.
\end{aligned}
\quad (19)
$$

Since $|b'| \leq \epsilon$, we have $b' + \epsilon \geq 0$ and hence $a + a' \leq a + a' + b' + \epsilon \leq \max\{a + a' + b' + \epsilon, b + a' + b' + \epsilon\}$. Similarly, $b + b' \leq \max\{a + a' + b' + \epsilon, b + a' + b' + \epsilon\}$ and finally we have $\max\{a + a', b + b'\} \leq \max\{a + a' + b' + \epsilon, b + a' + b' + \epsilon\}$. Clearly when $a' = b' = -\epsilon$, the equal sign holds.

**Proof for Theorem 1.** First, due to symmetry, optimizing (13) is equivalent to optimize

$$\mathbb{E}_{x \sim \mathcal{D}_1} \big[ \max_{\|\delta\|_\infty \leq \epsilon} (\max_{j \neq 1} f_{\boldsymbol{w}}(x + \delta)_j - f_{\boldsymbol{w}}(x + \delta)_1) \big] + \frac{\lambda}{2} \|\boldsymbol{w}\|_2^2. \tag{20}$$

Further, by Lemma 2 we can replace $\delta$ with its optimal value and transform the optimization objective above as

$$\mathbb{E}_{\hat{x} \sim \hat{\mathcal{D}}_1} (\max_{j \neq i} f_{\boldsymbol{w}}(\hat{x})_j - f_{\boldsymbol{w}}(\hat{x})_i) + \frac{\lambda}{2} \|\boldsymbol{w}\|_2^2, \tag{21}$$

where $\hat{\mathcal{D}}_1$ is the **adversarial data distribution**:

$$\hat{x}_{E,j} \sim \begin{cases} \mathcal{N}(\mu - \epsilon, \sigma^2), & j = 1 \\ \epsilon, & j \neq 1 \end{cases}, \quad \hat{x}_{C,j} \sim \begin{cases} \mathcal{N}(\mu - \epsilon, \sigma^2), & j \neq 1 \\ \epsilon, & j = 1 \end{cases}. \tag{22}$$

Now we calculate the expectation in (21).

$$\mathbb{E}_{\hat{x} \sim \hat{\mathcal{D}}_1} [(\max f_{\boldsymbol{w}}(\hat{x})_j - f_{\boldsymbol{w}}(\hat{x})_i)] + \frac{\lambda}{2} \|\boldsymbol{w}\|_2^2$$

$$= \mathbb{E}_{\hat{x} \sim \hat{\mathcal{D}}_1} [\max(w_1 \epsilon + w_2 \epsilon + w_2 \hat{x}_{C,3}, \ w_1 \epsilon + w_2 \epsilon + w_2 \hat{x}_{C,2} - w_1 \hat{x}_{E,1} - w_2(\hat{x}_{C,2} + \hat{x}_{C,3})] + \frac{\lambda}{2} \|\boldsymbol{w}\|_2^2$$

$$= \mathbb{E}_{\hat{x} \sim \hat{\mathcal{D}}_1} [w_1 \epsilon + w_2 \epsilon + w_2 \max(\hat{x}_{C,3}, \ \hat{x}_{C,2}) - w_1 \hat{x}_{E,1} - w_2(\hat{x}_{C,2} + \hat{x}_{C,3})] + \frac{\lambda}{2} \|\boldsymbol{w}\|_2^2$$

$$= w_1 \epsilon + w_2 \epsilon + w_2 \mathbb{E}_{\hat{x} \sim \hat{\mathcal{D}}_1} [\max(\hat{x}_{C,3}, \ \hat{x}_{C,2})] + \mathbb{E}_{\hat{x} \sim \hat{\mathcal{D}}_1} [-w_1 \hat{x}_{E,1} - w_2(\hat{x}_{C,2} + \hat{x}_{C,3})] + \frac{\lambda}{2} \|\boldsymbol{w}\|_2^2$$

$$= w_1 \epsilon + w_2 \epsilon + w_2 \mathbb{E}_{\hat{x} \sim \hat{\mathcal{D}}_1} [\max(\hat{x}_{C,3}, \ \hat{x}_{C,2})] + [-w_1(\mu - \epsilon) - 2 w_2(\mu - \epsilon)] + \frac{\lambda}{2} \|\boldsymbol{w}\|_2^2. \tag{23}$$

Finally, since $\hat{x}_{C,3}, \ \hat{x}_{C,2} \sim (\mu - \epsilon, \sigma^2)$ and they are independent, by Lemma 1 we have

$$\mathbb{E}[\max(\frac{\hat{x}_{C,3} - (\mu - \epsilon)}{\sigma}, \frac{\hat{x}_{C,2} - (\mu - \epsilon)}{\sigma})] = \frac{1}{\sqrt{\pi}}, \tag{24}$$

hence $\mathbb{E}_{\hat{x} \sim \hat{\mathcal{D}}_1} [\max(\hat{x}_{C,3}, \ \hat{x}_{C,2})] = \mu - \epsilon + \frac{\sigma}{\sqrt{\pi}}$.

Therefore, the optimizing objective can be simplified as

$$\mathcal{L}(f_{\boldsymbol{w}}) = (-\mu + 2\epsilon) w_1 + (-\mu + 2\epsilon + \frac{\sigma}{\sqrt{\pi}}) w_2 + \frac{\lambda}{2}(w_1^2 + w_2^2). \tag{25}$$

For $w_2$, we have

$$\frac{\partial \mathcal{L}}{\partial w_2} = -\mu + 2\epsilon + \frac{\sigma}{\sqrt{\pi}} + \lambda w_2. \tag{26}$$

Recall that $\sigma < \sqrt{\pi}\mu$. Let $\epsilon_0 = \frac{1}{2}(\mu - \frac{\sigma}{\sqrt{\pi}}) \in (0, \frac{\mu}{2})$. By analysing the sign of (26), it is clear that for $\epsilon \in (0, \epsilon_0)$, the optimal $w_2$ for minimizing the loss function (25) is

$$w_2 = \frac{\mu - 2\epsilon - \frac{\sigma}{\sqrt{\pi}}}{\lambda}. \tag{27}$$

However, for $\epsilon \in (\epsilon_0, \frac{\mu}{2})$, $\frac{\partial \mathcal{L}}{\partial w_2}$ is always negative, thus the returned $w_2$ by AT is $w_2 = 0$ under the constraint $w_2 \geq 0$.

By contrast,

$$\frac{\partial \mathcal{L}}{\partial w_1} = -\mu + 2\epsilon + \lambda w_1, \tag{28}$$

and for $\epsilon \in (0, \frac{\mu}{2})$, the optimal $w_1$ for minimizing the loss function (25) is always positive:

$$w_1 = \frac{\mu - 2\epsilon}{\lambda} > 0. \tag{29}$$

This ends our proof.

### E.3 PROOF FOR THEOREM 2

**Theorem 2** *For any $w_1 > 0$ and $\epsilon \in (0, \frac{\mu}{2})$, if $w_2 \in [0, w_1]$, a larger $w_2$ increases the possibility of the model distinguishing the adversarial examples from any other given class.*

To prove Theorem 2, we need the following lemma.

**Lemma 3** *Suppose that $X, Y \sim \mathcal{N}(1, \sigma_1^2)$ and they are independent, $\sigma_1 > 0$. Let $Z_t = X + tY$ where $t > 0$. Denote $u(t) = \Pr(Z_t > 0)$, then $u(t)$ is monotonically increasing at $t$ for $t \in [0, 1]$.*

*proof.* Note that $Z_t = X + tY \sim \mathcal{N}(1 + t, (1 + t^2)\sigma_1^2)$. Thus, the distribution function of $Z_t$ is $\Phi_t(z) = \Phi(\frac{z-1-t}{\sqrt{1+t^2}\sigma_1})$, and

$$u(t) = 1 - \Phi_t(0) = 1 - \Phi(\frac{-1-t}{\sqrt{1+t^2}\sigma_1}) = \Phi(\frac{1+t}{\sqrt{1+t^2}\sigma_1}),$$

$$u'(t) = p(\frac{1+t}{\sqrt{1+t^2}\sigma_1})\frac{\sqrt{1+t^2}\sigma_1 - (1+t)\frac{t\sigma_1}{\sqrt{1+t^2}}}{(1+t^2)\sigma_1^2} = p(\frac{1+t}{\sqrt{1+t^2}\sigma_1})\frac{(1+t^2) - (1+t)t}{(1+t^2)\sqrt{1+t^2}\sigma_1} \tag{30}$$

$$= p(\frac{1+t}{\sqrt{1+t^2}\sigma_1})\frac{1-t}{(1+t^2)\sqrt{1+t^2}\sigma_1}.$$

Therefore, for $t \in (0, 1)$, $u'(t) > 0$ and $u(t)$ is monotonically increasing at $t$ for $t \in [0, 1]$.

**Proof for Theorem 2.** Due to symmetry, it's suffice to show that given $w_1$, for $w_2 \in [0, w_1]$, the probability

$$\Pr(f_{\boldsymbol{w}}(\hat{x})_1 > f_{\boldsymbol{w}}(\hat{x})_2), \quad \hat{x} \sim \hat{\mathcal{D}}_1 \tag{31}$$

is monotonically increasing at $w_2$. Note that

$$f_{\boldsymbol{w}}(\hat{x})_1 - f_{\boldsymbol{w}}(\hat{x})_2 = w_1(\hat{x}_{E,1} - \hat{x}_{E,2}) + w_2(\hat{x}_{C,2} - \hat{x}_{C,1}),$$
$$\hat{x}_{E,1} - \hat{x}_{E,2} \sim \mathcal{N}(\mu - 2\epsilon, 2\sigma^2), \tag{32}$$
$$\hat{x}_{C,2} - \hat{x}_{C,1} \sim \mathcal{N}(\mu - 2\epsilon, 2\sigma^2).$$

By dividing $w_1 \cdot (\mu - 2\epsilon)$, and let $t = \frac{w_2}{w_1}$, $X = \frac{\hat{x}_{E,1} - \hat{x}_{E,2}}{\mu - 2\epsilon}$ and $Y = \frac{\hat{x}_{C,2} - \hat{x}_{C,1}}{\mu - 2\epsilon}$, from Lemma 3 we know that the probability

$$\Pr(f_{\boldsymbol{w}}(\hat{x})_1 - f_{\boldsymbol{w}}(\hat{x})_2 > 0) \tag{33}$$

is monotonically increasing at $t = \frac{w_2}{w_1}$, and hence increasing at $w_2$. This ends our proof.

### E.4 PROOF FOR THEOREM 3 AND COROLLARY 1

**Simplification of knowledge distillation as label smoothing.** In this context, the term 'symmetry' specifically refers to the symmetry of logits for the other two classes when taking the expectation in the loss function (equation 10). When considering data from class $y$, both the distribution of features $x_{E,i}$ and $x_{C_i}$ for the other two classes, as well as their respective weights $w_1$ and $w_2$, exhibit symmetry respectively. Consequently, after applying knowledge distillation, the expectation for logits of the other two classes in the objective loss function (equation 10) becomes identical. To simplify this process, we can employ label smoothing.

We prove Theorem 3 and Corollary 1 in the following. Recall that we define the robust loss under knowledge distillation as

$$\mathcal{L}_{\text{LS}}(f_{\boldsymbol{w}}) = \mathbb{E}_i\{\mathbb{E}_{x \sim \mathcal{D}_i}(1-\beta)[\max_{\|\delta\|_\infty \leq \epsilon}(\max_{j \neq i} f_{\boldsymbol{w}}(x+\delta)_j - f_{\boldsymbol{w}}(x+\delta)_i)] - \frac{\beta}{2}\sum_{j \neq i} f_{\boldsymbol{w}}(x+\delta)_j\} + \frac{\lambda}{2}\|\boldsymbol{w}\|_2^2.$$

$$\tag{34}$$

**Theorem 3** Consider AT with knowledge distillation loss (34). There exists an $\epsilon_1 > \epsilon_0$, such that for $\epsilon \in (0, \epsilon_1)$, the output function obtains $w_2 > 0$; for $\epsilon \in (\epsilon_1, \frac{1}{2}\mu)$, the output function returns $w_2 = 0$.

**Proof for Theorem 3.** Similar to the proof for Theorem 1, the optimization objective (34) can be simplified as

$$
\begin{aligned}
\mathcal{L}_{\text{LS}}(f_{\boldsymbol{w}}) &= (1-\beta)[(-\mu+2\epsilon)w_1 + (-\mu + 2\epsilon + \frac{\sigma}{\sqrt{\pi}})w_2] - \beta[\epsilon w_1 + \mu w_2] + \frac{\lambda}{2}(w_1^2 + w_2^2) \\
&= [(1-\beta)\mu + (2-3\beta)\epsilon]w_1 + [(1-\beta)(2\epsilon + \frac{\sigma}{\sqrt{\pi}}) - \mu]w_2 + \frac{\lambda}{2}(w_1^2 + w_2^2).
\end{aligned}
\tag{35}
$$

Thus

$$
\frac{\mathcal{L}_{\text{LS}}}{w_2} = (1-\beta)(2\epsilon + \frac{\sigma}{\sqrt{\pi}}) - \mu + \lambda w_2,
\tag{36}
$$

and let $\epsilon_1 = \frac{1}{2}(\frac{\mu}{1-\beta} - \frac{\sigma}{\sqrt{\pi}}) > \epsilon_0$, similar to the analysis for $\epsilon_0$, we have for $\epsilon \in (0, \epsilon_1)$, the output function obtains $w_2 > 0$; for $\epsilon \in (\epsilon_1, \frac{1}{2}\mu)$, the output function returns $w_2 = 0$. This ends our proof.

**Corollary 1** Let $w_2^*(\epsilon)$ be the value of $w_2$ returned by AT with (13), and $w_2^{\text{LS}}(\epsilon)$ be the value of $w_2$ returned by label smoothed loss (34). Then, for $\epsilon \in (0, \epsilon_1)$, we have $w_2^{\text{LS}}(\epsilon) > w_2^*(\epsilon)$.

**Proof for Corollary 1.** For $\epsilon \in (0, \epsilon_1)$, by analysing the sign of (36), we have

$$
w_2^{\text{LS}}(\epsilon) = \frac{\mu - (1-\beta)(2\epsilon + \frac{\sigma}{\sqrt{\pi}})}{\lambda},
\tag{37}
$$

and recall that in the proof for Theorem 1 we have

$$
w_2^*(\epsilon) = \frac{\mu - (2\epsilon + \frac{\sigma}{\sqrt{\pi}})}{\lambda},
\tag{38}
$$

thus it is clear that

$$
w_2^{\text{LS}}(\epsilon) - w_2^*(\epsilon) = \frac{\beta(2\epsilon + \frac{\sigma}{\sqrt{\pi}})}{\lambda} > 0.
\tag{39}
$$

This ends our proof.

# F   MORE EXPERIMENTS OF WAKE

We also conduct experiments comparing WAKE and baselines in other settings to further demonstrate its effectiveness in terms of improving robustness and mitigating robust overfitting.

## F.1   $\ell_2$-NORM AT

We conduct experiments on $\ell_2$-norm AT with $\epsilon = 128/255$ on CIFAR-10 dataset. The settings are the same as those of CIFAR-10 in Section 5.2. The results are shown in Table 3.

Table 3: Comparison of WAKE with vanilla AT and AT+KDSWA on $\ell_2$-norm.

| Dataset | Method | Robust Acc. (%) | | Clean Acc. (%) | |
|---|---|---|---|---|---|
| | | Best | Last | Best | Last |
| | AT | 67.3 | 64.5 | 88.6 | 88.7 |
| CIFAR-10 | AT + KDSWA | 68.9 | 68.3 | 89.4 | 89.7 |
| | AT + WAKE | **70.4** | **70.2** | **89.9** | **90.1** |

Results clearly show the advantage of WAKE over vanilla AT and KDSWA in terms of mitigating robust overfitting for $\ell_2$-norm AT.

## F.2 COMBINATION WITH OTHER METHODS

We conduct experiments on TRADES (Zhang et al., 2019) on CIFAR-10 dataset. The settings are the same as those of CIFAR-10 in Section 5.2. We follow the hyperparameters of TRADES from its original papers. The results are shown in Table 4.

Table 4: Comparison of WAKE with vanilla AT and AT+KDSWA combined with TRADES.

| Dataset | Method | Robust Acc. (%) | | Clean Acc. (%) | |
| --- | --- | --- | --- | --- | --- |
| | | Best | Last | Best | Last |
| | TRADES | 48.3 | 46.9 | 82.5 | 83.7 |
| CIFAR-10 | TRADES + KDSWA | 50.1 | 49.5 | 82.9 | 83.3 |
| | TRADES + WAKE | **50.7** | **50.4** | **83.8** | **84.1** |

Consistent with the results in the paper, these results clearly show that WAKE can also be combined with other advanced methods to further mitigate robust overfitting and improve adversarial robustness.

## F.3 TRANSFORMER ARCHITECTURE

We conduct experiments on transformer architecture DeiT-Ti (Touvron et al., 2021) on CIFAR-10 dataset. The settings are the same as those of CIFAR-10 in Section 5.2, and the robustness is evaluated using PGD-20. The results are shown in Table 5.

Table 5: Comparison of WAKE with vanilla AT and AT+KDSWA on DeiT-Ti architecture.

| Dataset | Method | Robust Acc. (%) | | Clean Acc. (%) | |
| --- | --- | --- | --- | --- | --- |
| | | Best | Last | Best | Last |
| | AT | 50.0 | 47.7 | 79.4 | 79.6 |
| CIFAR-10 | AT + KDSWA | 50.4 | 49.5 | 79.6 | 79.8 |
| | AT + WAKE | **50.6** | **50.3** | **80.1** | **80.4** |

Consistent with the results in the paper, these results clearly show that WAKE can also work on transformer architecture to further mitigate robust overfitting and improve adversarial robustness.

## G DETAILS FOR WAKE IMPLEMENTATION

### G.1 ALGORITHM FOR WAKE

We provide a detailed implementation algorithm of WAKE in Algrithm 2.

### G.2 DETAILS OF THE KNOWLEDGE DISTILLATION FUNCTION

Following Chen et al. (2021), the knowledge distillation (Hinton et al., 2015) function can be defined as:

$$\mathcal{KD}(f(\boldsymbol{\theta}_{\text{student}}; x), f(\boldsymbol{\theta}_{\text{teacher}}; x)) = \text{KL}[\text{softmax}(\frac{f(\boldsymbol{\theta}_{\text{student}}; x)}{T}), \text{softmax}(\frac{f(\boldsymbol{\theta}_{\text{teacher}}; x)}{T})], \quad (40)$$

where $\text{KL}(\cdot, \cdot)$ is the Kullback-Leibler divergence and $T$ is the distillation temperature.

## H ADDITIONAL RELATED WORK

---

**Algorithm 2: W**eight **A**verage guided **K**nowledg**E** Distillation (WAKE)

---

**Input:** A DNN classifier $f_{\boldsymbol{\theta}}(\cdot)$ with parameter $\boldsymbol{\theta}$; Train dataset $D = \{(x_i, y_i)\}_{i=1}^{N}$; Batch size $m$;
        Initial perturbation margin $\epsilon$; Train epochs $N$; Learning rate $\eta$; Weight average decay rate
        $\alpha$; Knowledge distillation warm-up start epoch $t_s$ and end epoch $t_e$; hyper-parameter $\lambda$.
**Output:** A robust classifier $\bar{f}_{\bar{\boldsymbol{\theta}}}$ with less overfitting

**for** $t \leftarrow 1, 2, \cdots, T$ **do**
     **for** *Every minibatch (x,y) in trainset $D$* **do**
         $\delta \leftarrow \max_{\|\delta\|_p \leq \epsilon} \ell_{\text{CE}}(f(\boldsymbol{\theta}, x + \delta), y)$;
         **if** $t > t_s$ **then**
            $\tilde{y} \leftarrow f(\bar{\boldsymbol{\theta}}, x + \delta)$(stop gradient);
            **if** $t < t_e$ **then**
               $\lambda_t \leftarrow \frac{t - t_s}{t_e - t_s} \cdot \lambda$;
            **else**
               $\lambda_t \leftarrow 1$;
            $\boldsymbol{\theta} \leftarrow \boldsymbol{\theta} - \eta \nabla_{\boldsymbol{\theta}}[(1 - \lambda_t)\ell_{\text{CE}}(f(\boldsymbol{\theta}, x + \delta), y) + \lambda_t \cdot \mathcal{KD}(f(\boldsymbol{\theta}; x + \delta), \tilde{y}))].$;
         **else**
            $\boldsymbol{\theta} \leftarrow \boldsymbol{\theta} - \eta \nabla_{\boldsymbol{\theta}}[\ell_{\text{CE}}(f(\boldsymbol{\theta}, x + \delta), y)]$;
         **if** $t < t_e$ **then**
            $\bar{\boldsymbol{\theta}} \leftarrow \alpha\bar{\boldsymbol{\theta}} + (1 - \alpha)\boldsymbol{\theta}$;

**return** $f_{\bar{\boldsymbol{\theta}}}$;

---

## H.1 ADVERSARIAL TRAINING AND ADVERSARIAL ROBUSTNESS

The adversarial robustness and adversarial training has become popular research topic since the discovery of adversarial examples (Szegedy et al., 2013; Goodfellow et al., 2014), which uncovers that DNNs can be easily fooled to make wrong decisions by adversarial examples that are crafted by adding small perturbations to normal examples. The malicious adversaries can conduct adversarial attacks by crafting adversarial examples, which cause serious safety concerns regarding the deployment of DNNs. Following this discovery, various types of adversarial attack methods have been proposed, including gradient-based (Carlini & Wagner, 2017b; Liu et al., 2022), query-based (Andriushchenko & Flammarion, 2020; Bai et al., 2019), decision-based (Brendel et al., 2017; Chen et al., 2020) and demonstration-based (Wang et al., 2023; Wei et al., 2023b) attacks on various models and tasks.

In response to such adversarial threats, numerous defense approaches have also been proposed, such as adversarial example detection (Grosse et al., 2017; Tian et al., 2018) and purification (Bai et al., 2019; Nie et al., 2022), parameter regularization (Jakubovitz & Giryes, 2018; Wei et al., 2023c), randomized smoothing (Cohen et al., 2019; Levine & Feizi, 2020), among which adversarial training methods (Madry et al., 2017; Wang et al., 2019) has been considered as the most promising defending method against adversarial attacks (Carlini & Wagner, 2017a; Athalye et al., 2018). Through this research thread, there are also other perspectives on improving adversarial training, including architecture design (Huang et al., 2021; Mo et al., 2022), data augmentation (Rebuffi et al., 2021b;a), optimization objective design (Wang et al., 2020; Pang et al., 2022).

## H.2 ADVERSARIAL ROBUSTNESS DISTILLATION

Besides adversarial training, there are also several papers on distilling adversarial robustness from teacher models (Goldblum et al., 2020; Zhu et al., 2021; Zi et al., 2021; Huang et al., 2023; Yue et al., 2023). Similar to conventional knowledge distillation (Hinton et al., 2015; Gou et al., 2021), this thread works toward training an adversarially robust student model with a robust teacher model. By designing proper distilling objectives and algorithms, these works can enhance the robustness of the trained student model.

There are several differences between our proposed WAKE method and these adversarial robustness distillation methods. First, WAKE is designed to mitigate robust overfitting in adversarial training, which is different from existing work typically for improving adversarial robustness. To the best of

our knowledge, the KDSWA (Chen et al., 2021) is the only existing distillation method designed for the same purpose, thus we only include KDSWA and the vanilla adversarial training method as baselines in experiments. Moreover, WAKE uses the weight-averaged model as the teacher model, which does not require a given robust teacher model. Therefore, not only WAKE can save large amounts of computational resources, but also its robustness is not dependent on another teacher model.

