# OpenReview forum: "Characterizing Robust Overfitting in Adversarial Training via Cross-Class Features"
_ICLR.cc/2024/Conference — Submitted to ICLR 2024_

### Official Review · Reviewer_iycg · 2023-10-30

**Soundness:** 3 good
**Presentation:** 3 good
**Contribution:** 3 good
**Rating:** 5
**Confidence:** 5

**Summary:**

In this paper, the authors primarily concentrate on addressing the robust overfitting problem. In pursuit of this objective, they identify intriguing properties within both cross-class and class-specific features. Then, both empirical and theoretical evidence show that knowledge distillation helps mitigate robust overﬁtting by preserving the observed properties. The experiments further verify this.

**Strengths:**

1. Understanding robust overfitting via cross-class and class-specific features is interesting.
1. The empirical and theoretical evidence could support the recent common sense that knowledge distillation is beneficial to mitigate robust overfitting.
1. The experiments show that a weight-average guided knowledge distillation method can boost robustness.

**Weaknesses:**

1. In Eq. 10, the authors use label smoothing to simulate knowledge distillation. I am wondering if this simulation is reasonable. The authors only said that ''due to symmetry'', but it is difficult to understand.

1. On page 6, the authors claimed that "... derive a large enough logit" and "... Since the final checkpoint makes decisions based only on these limited features, the logit for the correct class is not large enough". These two conclusions are very difficult to understand because I find it challenging to infer them from the existing experiments. Figure 4 only shows the correlation matrix which cannot reflect the logit information.

1. In Equation 7, why do the class-specific or cross-class features only appear in robust features?

1. The results shown in Section 4.2 are somehow weak I think. For example, it is difficult to see that feature $x_{E,i}$ will dominate the whole training objective when $w_1>0$.

1. There are many adversarial robust distillation methods in previous works, so it's better to compare them in details like [a].

[a] Revisiting Adversarial Robustness Distillation: Robust Soft Labels Make Student Better. ICCV 2021.

**Questions:**

Apart from the issues mentioned in the Weaknesses section, there is another concern: this paper mainly focuses on robust overfitting. How about catastrophic overfitting in Fast-AT?

---

> ### Author Response · Authors · 2023-11-20
> **Response to Reviewer iycg (part 1/2)**
>
> Thank you for your valuable feedback. We have revised our manuscript based on your comments and address your concerns as follows.
>
> ---
>
> **Q1**: In Eq. 10, the authors use label smoothing to simulate knowledge distillation. I am wondering if this simulation is reasonable. The authors only said that ''due to symmetry'', but it is difficult to understand.
>
> **A1**: Thanks for pointing that out. In this context, the term 'symmetry' specifically refers to the symmetry of logits for the other two classes when taking the expectation in the loss function (equation 10). When considering data from class $y$, both the distribution of features $x_{E,i}$  and $x_{C_i}$ for the other two classes, as well as their respective weights $w_1$ and $w_2$, exhibit symmetry respectively. Consequently, after applying knowledge distillation, the expectation for logits of the other two classes in the objective loss function (equation 10) becomes identical. To simplify this process, we can employ label smoothing.
>
> We have incorporated this discussion into our revision.
>
> ---
>
> **Q2**: On page 6, the authors claimed that "... derive a large enough logit" and "... Since the final checkpoint makes decisions based only on these limited features, the logit for the correct class is not large enough". These two conclusions are very difficult to understand because I find it challenging to infer them from the existing experiments. Figure 4 only shows the correlation matrix which cannot reflect the logit information.
>
> **A2**: Thank you for your careful reading. We have toned down such claims in the updated paper as
>
> > Since the final checkpoint makes decisions based only on these limited features, it fails to leverage comprehensive features for classification, making the model more vulnerable to adversarial attacks on these samples.
>
> ---
>
> **Q3**: In Equation 7, why do the class-specific or cross-class features only appear in robust features?
>
> **A3**: Thanks for your careful reading. We want to emphasize that our claim does not assert that features of this nature exclusively appear in robust features. As clarified earlier, our data model, inspired by [1], primarily centers on the inherent relationship between these two types of robust features, and non-robust features do not affect our main claims. To illustrate, consider a non-robust, class-specific feature $x_n$ with a Gaussian mean $\eta$ (as introduced in [1]), where $\eta$ is a small positive value. For adversarial training with perturbation bound $\epsilon>\eta$ (generally holds since $\eta$ is small), the weight $w_n$ for $x_n$ will be set to 0, as demonstrated by the proof of Theorem 1. Consequently, under adversarial training, $x_n$ does not exert an influence on the decision logic. This same conclusion extends to cross-class features as well.
>
> [1] Robustness may be at odds with accuracy. ICLR 2019
>
> ---
>
> **Q4**: The results shown in Section 4.2 are somehow weak I think. For example, it is difficult to see that feature $x_{E,i}$ will dominate the whole training objective when $w_1>0$.
>
> **A4**: Thank you for your valuable insights. We would like to clarify that the assumption regarding the dominance of $x_{E,i}$ is primarily supported by our empirical findings, which reveal a gradual forgetting of cross-class features $x_{C,i}$ and an increasing dominance of class-specific features $x_{E,i}$ during adversarial training (AT). These observed tendencies of these features during AT lend credibility to the reasonable supposition that feature $x_{E,i}$ will indeed dominate the training objective.

---

> ### Author Response · Authors · 2023-11-20
> **Response to Reviewer iycg (part 2/2)**
>
> **Q5**: There are many adversarial robust distillation methods in previous works, so it's better to compare them in details like [a].
>
> **A5**: Thank you for your thoughtful suggestions. The primary distinction between our method and adversarial robust distillation methods lies in their respective purposes and settings. Our method is specifically designed to address robust overfitting, whereas robust distillation methods are focused on distilling adversarial robustness from pre-trained teacher models. Furthermore, it's noteworthy that our proposed method is entirely trained from scratch, in contrast to robust distillation methods that require a pre-trained teacher.
>
> ---
>
> **Q6**: This paper mainly focuses on robust overfitting. How about catastrophic overfitting in Fast-AT?
>
> **A6**: Thank you for offering this valuable perspective. It's essential to note that the majority of existing papers [2-4] on understanding robust overfitting do not explicitly delve into the topic of catastrophic overfitting in the context of Fast-AT, which indeed remains an open research area.
>
> In line with your suggestion, we have extended our investigations to include Fast-AT for the CIFAR-10 dataset, employing an $\ell_\infty$-norm perturbation bound of $\epsilon=8/255$. The detailed results are presented in **Appendix D.6**. Notably, after catastrophic overfitting, there is a significant reduction in the usage of cross-class features. This observation aligns with our understanding, indicating that the model also tends to forget cross-class features after experiencing catastrophic overfitting.
>
> ---
>
> [a] Revisiting Adversarial Robustness Distillation: Robust Soft Labels Make Student Better. ICCV 2021
>
> [2] Understanding Robust Overfitting of Adversarial Training and Beyond. ICML 2022
>
> [3] On the Onset of Robust Overfitting in Adversarial Training. arxiv preprint
>
> [4] Understanding and combating robust overfitting via input loss landscape analysis and regularization. Pattern Recognition

---

> ### Author Response · Authors · 2023-11-22
> **Could you please have a look at our rebuttal?**
>
> Dear Reviewer iycg, thanks for your time reviewing our paper. We have meticulously prepared a detailed response addressing your concerns and revised our paper accordingly. Could you please have a look to see if there are further questions? Your invaluable input is greatly appreciated. Thank you once again, and we hope you have a wonderful day!

---

### Official Review · Reviewer_quy6 · 2023-10-31

**Soundness:** 2 fair
**Presentation:** 2 fair
**Contribution:** 2 fair
**Rating:** 3
**Confidence:** 4

**Summary:**

As claimed in this paper, robust overfitting, which although has been widely discussed, has not been fully understood. In this paper a new interpretation in the view of feature attribution is proposed and then a follow-up method is designed.

**Strengths:**

Robust overfitting is an interesting phenomena happened in adversarial training. Investigating it in the view of feature attribution may bring new understanding for it.

**Weaknesses:**

- Robust overfitting is specific to adversarial training. Thus, the discussion should be emphasized more on special properties of adversarial attack/examples/training. However, the idea given in this paper fails in this aspect: in regular training, it is also the case that the model first learn cross class information and then focusing on a specific model, e.g., the well-known neural collapse saying that the features of examples of one class will collapse to a direction.

- The unclearness to the regular training also could be observed in their proposed method: stochastic weight averaging (SWA) is a standard technique to enhance generalization capability in regular training. In the current experiment, it is hard to say the advantage of the proposed WAKE comes from SWA or it comes specific property of adversarial training, then now it cannot well support the main contribution.

- Another weakness is that the metric used to support the conclusion is not very convincing. For example, it is believed that features in different layers capture different information, e.g., cross-class or in-class information. Simply put all features together seems too weak.

**Questions:**

- I want to see specific properties in adversarial training. Thus, could I see the performance of the same experiment but on regular training?

- There are many metric that can measure the information about class and examples learned by DNNs. Could the authors use other metric and also observe the similar phoneme as measured by CAS？

---

> ### Author Response · Authors · 2023-11-20
> **Response to Reviewer quy6**
>
> Thank you for your detailed and valuable feedback. We have revised our manuscript based on your comments and addressed your concerns as follows.
>
> ---
>
> **Q1**: Robust overfitting is specific to adversarial training. Thus, the discussion should be emphasized more on special properties of adversarial attack/examples/training. However, the idea given in this paper fails in this aspect: in regular training, it is also the case that the model first learn cross class information and then focusing on a specific model, e.g., the well-known neural collapse saying that the features of examples of one class will collapse to a direction.
>
> **A1**: Thank you for reminding this viewpoint.
>
> First, we want to emphasize that we have indeed taken into account the distinction in the explanation for robust overfitting between adversarial training and regular training. In Section 3.3 and illustrated in Figure 3, our experiments encompass different values of $\epsilon$, with a small $\epsilon$ (2/255) representing a scenario closer to regular training. Gradually increasing $\epsilon$ can be seen as a transition towards more rigorous adversarial training. Notably, the significance of cross-class feature forgetting becomes more pronounced as $\epsilon$ increases. This observation aligns with the phenomenon that robust overfitting is uniquely evident in adversarial training scenarios.
>
> Following your suggestions, we have extended our experimental scope to include regular training on the CIFAR-10 dataset. The experimental settings mirror those outlined in Section 5, with the sole distinction being the absence of perturbations in regular training. The results are presented in **Appendix D.5**. Specifically, considering that regular training prioritizes natural generalization and exhibits minimal robustness, we have calculated the feature attribution vectors **using clean examples**. These vectors were computed for epochs $\\{50, 100, 150, 200\\}$. Notably, the results reveal a lack of clear differences between them, particularly in the latter stages (150th and 200th), where the training tends to converge. This observation is consistent with the characteristic of regular training, which typically does not exhibit overfitting.
>
> ---
>
> **Q2**: The unclearness to the regular training also could be observed in their proposed method: stochastic weight averaging (SWA) is a standard technique to enhance generalization capability in regular training. In the current experiment, it is hard to say the advantage of the proposed WAKE comes from SWA or it comes specific property of adversarial training, then now it cannot well support the main contribution.
>
> **A2**: Thank you for pointing this out. It's essential to recall that in our comparison with the baseline KDWSA [1], which combines knowledge distillation (KD) and stochastic weight averaging (SWA), the performance of WAKE is notably superior to KDSWA, not to mention using SWA alone. To provide further clarification, we have added a table below for a direct comparison. The experimental settings are identical to those in Table 1 for the CIFAR-10 dataset. Consequently, it is clear that the improvement achieved by WAKE is not solely attributed to SWA; rather, it stems from the specific properties of adversarial training.
>
> | Method | Robust Acc (Best) | Robust Acc (Last) |
> | --- | --- | --- |
> | AT+KD | 48.6 | 46.0 |
> | AT+SWA | 49.2 | 47.1 |
> | AT+KDSWA | 49.8 | 49.6 |
> | AT+WAKE | 50.4 | 50.1 |
>
> ---
>
> **Q3**: Another weakness is that the metric used to support the conclusion is not very convincing. For example, it is believed that features in different layers capture different information, e.g., cross-class or in-class information. Simply put all features together seems too weak.
>
> **A3**: Thanks for your kind suggestion. Firstly, it's important to clarify that CAS does not aggregate all features. Instead, CAS exclusively utilizes the features from the penultimate layer, just before the linear classification head. This selection is based on the understanding that features in this layer hold the highest influence on the classification logits, providing a clear reflection of the decision-making process.
>
> **Q4**: There are many metric that can measure the information about class and examples learned by DNNs. Could the authors use other metric and also observe the similar phoneme as measured by CAS？
>
> **A4**: We appreciate your suggestion regarding alternative metrics to quantify the usage of cross-class features, but we do not know which metric you are referring. Could you clarify which specific metric you are referring? This will assist us in addressing your suggestion more accurately.
>
> ---
>
> [1] Robust overfitting may be mitigated by properly learned smoothening. ICLR 2021
>
> ---
>
> We truly appreciate your valuable and detailed feedback. If you have any further questions or concerns, please let us know.

---

> > ### Comment · Reviewer_quy6 · 2023-11-21
> > **thanks for the explanation**
> >
> > Thanks for the authors' response, especially the clarification on which features are used and the direct comparison. As the supplementary table shows, the proposed WAKE is better than others, however not significant, and more importantly the improvement does not come from decreasing the generalization gap. I think all the reviewers have consistent concerns about the motivation, the algorithm, and the experiments. So I would like to keep my score.

---

> > > ### Author Response · Authors · 2023-11-22
> > > **Please clarify your concern**
> > >
> > > Dear Reviewer quy6,
> > >
> > > Thank you for your prompt response and acknowledging the performance of WAKE is better than others. We further address your concern as follows.
> > >
> > > - The proposed WAKE is better than others, however not significant.
> > >     - Contrary to the notion of "not significant", our method, WAKE, actually exhibits a noteworthy enhancement over KDSWA in several settings. For example, referencing the $\ell_2$-norm results in the Appendix, the robust accuracy of WAKE is about 2% higher than KDSWA in both of the best and last cases. This margin is substantial in the context of the adversarial domain.
> > >     - Beyond just robustness improvements, an added advantage of WAKE is its superior computational efficiency. While KDSWA necessitates the pre-training of teacher models, WAKE leverages a weight-averaged model as the teacher, streamlining the process and reducing computational demands.
> > >     - In addition to our justification regarding the significance of the improvement, we would like to further emphasize that the main contribution of this paper extends beyond the newly proposed method. A significant aspect of our work lies in the novel interpretation of robust overfitting from the feature attribution perspective. We believe this interpretation offers valuable insights that go beyond the specific details of the proposed method
> > > - Improvement does not come from decreasing the generalization gap.
> > >     - We would like to provide further clarification that the observed improvement does indeed stem from a reduction in the generalization gap. As evident in the table below, the generalization gap of our method is smaller to that of the baseline methods. This emphasizes that the enhancements achieved by our proposed method are specifically targeted at mitigating the generalization gap rather than solely boosting robustness.
> > >
> > >     | Method | Train Robust Acc (Last) | Test Robust Acc (Last) | Generalization Gap |
> > >     | --- | --- | --- | --- |
> > >     | AT | 80.6% | 42.5% | 38.1% |
> > >     | AT+KDSWA | 72.4% | 49.6% | 22.8% |
> > >     | AT+WAKE | 71.6% | **50.1%** | **21.5%** |
> > > - All the reviewers have consistent concerns about the motivation, the algorithm, and the experiments.
> > >     - Firstly, it is crucial to highlight that these concerns raised by you do not appear to be shared by other reviewers. For instance, reviewer **fn47** finds our proposed understanding is **intuitively clear and is backed well by empirical results**, and reviewer **iycg** believes **the empirical and theoretical evidence could support the recent common sense that knowledge distillation is beneficial to mitigate robust overfitting**. Neither of them has expressed concerns regarding the motivation, algorithm, or experiments, or if they had, those concerns were effectively addressed in our rebuttal.
> > >     - Furthermore, upon careful consideration of your comments, we have not identified any specific weaknesses or questions related to the motivation, the algorithm, or the experiments that have not already been addressed in our response. Could you please clarify your further concerns on these aspects? Your additional insights will be instrumental in refining our work to meet the highest standards.
> > > ---
> > > We truly appreciate your valuable and detailed feedback. We hope this further clarification addresses your concerns and look forward to your feedback.

---

> > > ### Author Response · Authors · 2023-11-23
> > > **Could you please have a look at our further response?**
> > >
> > > Dear reviewer quy6,
> > >
> > > Thank you once again for your feedback. We have prepared a detailed response to address your additional concerns. Could you please take a moment to review it and let us know if you have any further questions? Your valuable input is greatly appreciated. Thank you once again, and we hope you have a nice day!

---

### Official Review · Reviewer_NSUP · 2023-10-31

**Soundness:** 2 fair
**Presentation:** 2 fair
**Contribution:** 2 fair
**Rating:** 3
**Confidence:** 5

**Summary:**

This paper studies the effects of cross-class features and specific-class features to the performance of adversarial training. The conclusion is that the cross-class features are more robust and important to the robust accuracy.

**Strengths:**

- The empirical finding is interesting and possibly helpful in practice to improve the performance of adversarial training.

**Weaknesses:**

- The theories developed are very restrictive and under extremely unrealistic assumptions
   -  Data have 6 features and each features follow a Gaussian with mean zero.
   -  It only considers a linear model with two parameters $w_1$ for the exclusive feature and $w_2$ for the shared feature.

- The theoretical results obtained are humble and do not really reflect the empirical finding. Moreover, the statement of Theorem 2 is ambiguous. It is not clear what it does mean by "a larger w2 increases the possibility of the model distinguishing the adversarial examples from any other given class".

- When taking average of feature vectors for a class to visualize the matrix $C$ and CAS, this cannot reflect the tendency/behavior of each feature vector. To me, a better way is to compute and visualize instance-wisely.

- The proposed practical method lacks novelty and does not have a strong connection to theory developed and experiments are humble.

**Questions:**

What does it mean by "a larger w2 increases the possibility of the model distinguishing the adversarial examples from any other given class" in Theorem 2?

Can you compute and visualize C and CAS instance-wisely? For example, do it for a pair of feature vectors in class i and class j, and then take average.

---

> ### Author Response · Authors · 2023-11-20
> **Response to Reviewer NSUP (1/2)**
>
> Thank you for your detailed and valuable feedback. We have revised our manuscript based on your comments and addressed your concerns as follows.
>
> ---
>
> **Q1**: The theories developed are very restrictive and under extremely unrealistic assumptions.
>
>   - Data have 6 features and each features follow a Gaussian with mean zero.
>   - It only considers a linear model with two parameters $w_1$ for the exclusive feature and $w_2$ for the shared feature.
>
> **A1**: We appreciate your feedback, but we have to point out this is an unjustified criticism to our paper.
>
> - Firstly, our choice of data distribution is rooted in a well-established model from [1], which separates robust and non-robust features through similar Gaussian distributions. This model has been widely adopted in subsequent studies, such as [2], for the analysis of adversarial robustness.
> - Moreover, though [1] considers $d$ non-robust features, they simplified the linear function due to symmetry and they only use **2 parameters** (one for robust features and another for non-robust features) in the **linear model**. Therefore, we believe modeling the **6 features** among 3 classes is sufficient to analyze the relationship between cross-class features and adversarial robustness.
> - Additionally, in the realm of adversarial robustness, many theoretical analyses, including but clearly not limited to [3,4,5], are also based on **linear models.** It is worth noting that linear models are sufficient to convey valuable insights, and we don't perceive the analysis within a linear model as a weakness of our paper.
>
> ---
>
> **Q2**: The theoretical results obtained are humble and do not really reflect the empirical finding. Moreover, the statement of Theorem 2 is ambiguous. It is not clear what it does mean by "a larger w2 increases the possibility of the model distinguishing the adversarial examples from any other given class".
>
> **A2**: We appreciate your feedback, but the theoretical results are well aligned with the empirical findings.
>
> - Theorem 1 explains why larger $\epsilon$ in adversarial training forgets more cross-class features by showing cross-class features are more sensitive to robust loss in adversarial training, which is consistent with Figure 3.
> - Theorem 2 explains why forgetting cross-class features leads to a decrease of robustness by showing larger weight of cross-class features $w_2$ can bring better robustness, which is consistent with the overall finding of this paper.
> - Theorem 3 and Corollary 1 show how knowledge distillation mitigates robust overfitting which is consistent with Figure 4(b)(c).
>
> As for theorem 2, following your feedback, we have refined the claim in Theorem 2 to make it clearer in our revised paper, and we copied it below:
>
> >  For any class $y$, consider weights $w_1 > 0$, $w_2 \in [0, w_1]$, and $\epsilon \in (0, \frac{\mu}{2})$. When sampling $x$ from the distribution of class $y$, increasing the value of $w_2$ enhances the possibility of the model assigning a higher logit to class $y$ than to any other classes $y'\ne y$ under adversarial attack. In other words, the probability $\Pr_{x\sim\mathcal D_y}[f_w(x+\delta))\_{y}>f_w(x+\delta)\_{y'},\forall \delta:\|\delta\|_\infty\le\epsilon]$ monotonically increases with $w_2$ within the range $[0, w_1]$.
>
>
>
> ---
>
> [1] Robustness may be at odds with accuracy. ICLR 2019
>
> [2] Adversarial Examples Are Not Bugs, They Are Features. NeurIPS 2019
>
> [3] Theoretically Principled Trade-off between Robustness and Accuracy. ICML 2019
>
> [4] To be Robust or to be Fair: Towards Fairness in Adversarial Training. ICML 2020
>
> [5] On the Tradeoff Between Robustness and Fairness. NeurIPS 2023

---

> > ### Comment · Reviewer_NSUP · 2023-11-22
> > **Thanks for your feedback**
> >
> > Thanks the authors for answers my questions. Currently, I am keen on to keep my score. The reasons include:
> > - I am not very convincing of using $A_i(x)$ to represent the features in $g(x)$ that characterize the class $i$. I agree that the if $x$ has the class $i$ and the classifier correctly predicts $x$, many value $W_{ij}g(x)_j$ receive remarkable values. Moreover, to compute $A^y$, you attack the test examples $x$ in the class $y$ to gain $x^a$, evaluate the attribution vector of $x^a$ for the class $y$ (i.e., $A_y(x^a)$), and take average of them to yield $A^y$. My concern is that after attacking, $x^a$ has the predicted label $y^a \neq y$, hence if $A_y(x^a)$ makes sense in this context. Moreover, you take average of all $A_y(x^a)$ where $x^a$ can be attacked to different labels. It is hard for me to interpret the meaning of this average.
> > -  I cannot classify this work to a work with strong contributions to theory or proposing a novel practical method. With respect to the first perspective, the assumptions made in this work are super-unrealistic to me, while the proposed practical method is limited in novelty with just small improvements comparing to AT + KDSWA.

---

> > > ### Author Response · Authors · 2023-11-23
> > > **Further response to reviewer NSUP**
> > >
> > > Dear reviewer NSUP,
> > >
> > > Thanks for your prompt response and valuable feedback, we really appreciate it. We further address your concerns below.
> > >
> > > ---
> > > - After attacking, $x^a$ has the predicted label $y^a\ne y$, hence if $A_y(x^a)$ makes sense in this context.
> > >     - Thanks for considering this carefully. We justify the use of the provided image for visualization based on the following reasons. In this context, our focus is on the attribution of the features, specifically, which features are used for classification in adversarial examples. To characterize the robust features [1], we need to utilize adversarial attacks to create them. Even if the image is misclassified into different classes, it is still valid for characterizing which features are used, rather than which class is predicted. This aligns with explainability methods like Grad-CAM [2], where the emphasis is on feature attribution for a specific class, rather than the predicted class. Such methods remain meaningful even when the model misclassifies a natural sample into other classes.
> > > ---
> > > - Moreover, you take average of all  where  can be attacked to different labels.
> > >     - Thank you once again for mentioning this point. As we have clarified, we choose to use vectors instead of classes because we believe that averaging over samples can provide a more comprehensive and holistic understanding of the attribution for a specific class. Simply averaging over all pairs for two classes results in a large variance, which does not provide any information about the use of cross-class features.
> > >     - Additionally, we have calculated feature attribution correlation matrices on an instance-wise basis for both $\ell_\infty$ and $\ell_2$-norm adversarial training in Appendix D.4. The results and conclusions remain the same, which does not affect our claims.
> > > ---
> > > - I cannot classify this work to a work with strong contributions to theory or proposing a novel practical method. With respect to the first perspective, the assumptions made in this work are super unrealistic to me, while the proposed practical method is limited in novelty with just small improvements comparing to AT + KDSWA.
> > >
> > >    - Thank you for carefully considering the contribution of this work. As mentioned in Section 1, the main contribution of this work is the novel interpretation of robust overfitting, which is strongly supported by our extensive empirical evidence. We want to clarify that our paper is not intended to be a theory paper. The study of adversarial robustness is a predominantly empirical field, and many papers [3,4,5] that aim to understand robust overfitting do not provide theoretical analysis.
> > >
> > >   - Furthermore, regarding our proposed WAKE, we claim that WAKE actually exhibits a noteworthy enhancement over KDSWA in several settings. For example, referencing the $\ell_2$-norm results in the Appendix, the robust accuracy of WAKE is about 2% higher than KDSWA in both the best and last cases. This margin is substantial in the context of the adversarial domain. Beyond just robustness improvements, an added advantage of WAKE is its superior computational efficiency. While KDSWA necessitates the pre-training of teacher models, WAKE leverages a weight-averaged model as the teacher, streamlining the process and reducing computational demands.
> > >
> > >     ---
> > >
> > >
> > > We truly appreciate your valuable and detailed feedback. We hope this further clarification addresses your concerns and look forward to your feedback.
> > >
> > > [1] Adversarial Examples Are Not Bugs, They Are Features. NeurIPS 2019
> > >
> > > [2] Grad-CAM: Visual Explanations from Deep Networks via Gradient-based Localization. ICCV 2017
> > >
> > > [3] Understanding Robust Overfitting of Adversarial Training and Beyond. ICML 2022
> > >
> > > [4] On the Onset of Robust Overfitting in Adversarial Training. arxiv preprint
> > >
> > > [5] Understanding and combating robust overfitting via input loss landscape analysis and regularization. Pattern Recognition

---

> > > ### Author Response · Authors · 2023-11-23
> > > **Could you please have a look at our further response?**
> > >
> > > Dear reviewer NSUP,
> > >
> > > Thank you once again for your valuable comments. We have prepared a detailed response to address your additional concerns. Could you please take a moment to review it and let us know if you have any further questions? Your valuable input is greatly appreciated. Thank you once again, and wish you all the best!

---

> ### Author Response · Authors · 2023-11-20
> **Response to Reviewer NSUP (2/2)**
>
> **Q3**: When taking average of feature vectors for a class to visualize the matrix and CAS, this cannot reflect the tendency/behavior of each feature vector. To me, a better way is to compute and visualize instance-wisely.
>
> **A3**: Thank you for your thoughtful suggestion. Our choice of using vectors over classes is based on the belief that averaging over samples can yield a more comprehensive and holistic understanding of the attribution for different classes.
>
> Following your suggestion, we additionally calculated feature attribution correlation matrices on an instance-wise basis for both $\ell_\infty$ and $\ell_2$ adversarial training. The results are presented in **Appendix D.4**. To elaborate, when considering classes $i$ and $j$, for each sample $x$ from class $i$, we identify its most similar sample $x'$ from class $j$.  We then calculate their cosine similarity and average the results over all samples in class $i$.
>
> In this context, $x'$ can be interpreted as the sample in class $j$ that shares the most cross-class features with $x$ among all samples in class $j$. This metric provides a meaningful way to quantify the utilization of cross-class features. We did attempt to average over all sample pairs $(x, x')$ in classes $i$ and $j$, but due to high variance among samples, each element in the correlation matrix $C$ hovered near 0 throughout all epochs in adversarial training, rendering it unable to provide meaningful information.
>
> Consistent with the results for class-wise attribution vectors, it is still observed that there is a significant decrease in the usage of cross-class features from the best checkpoint to the last for both $\ell_\infty$ and $\ell_2$-AT. This observation further substantiates our understanding of robust overfitting.
>
> ---
>
> **Q4**: The proposed practical method lacks novelty and does not have a strong connection to theory developed and experiments are humble
>
> **A4**:
>
> - In terms of novelty, it's crucial to highlight a key distinction of our method from existing distillation methods. Unlike approaches like KDWSA, which necessitates a pre-trained teacher model, our proposed method, WAKE, stands out in its efficiency and ease of implementation. WAKE does not rely on a pre-trained teacher model, making it a more straightforward and resource-efficient solution. Regarding your criticism on novelty, could you kindly specify any existing works that significantly overlap with our proposed WAKE?
> - In terms of connection to theory, in this paper, we showed the importance of cross-class features for improving adversarial robustness, and the effectiveness of knowledge distillation to preserve such features. Thus, a natural solution is finding a better distillation teacher model for more precise cross-class feature information. This shows a clear connection between our theory and the proposed method.
> - In terms of experiments, we also conducted comprehensive experiments which are shown in the Appendix. First, we delved into empirical investigations to uncover the relationship between cross-class features and robust overfitting (as detailed in Appendix D). Additionally, we conducted a series of experiments for WAKE (as detailed in Appendix F). These encompassed examinations involving $\ell_2$-norm, synergies with other adversarial training methods, and the exploration of alternative model architectures.
>
> ---
>
> We truly appreciate your valuable and detailed feedback. If you have any further questions or concerns, please let us know.

---

> ### Author Response · Authors · 2023-11-22
> **Could you please have a look at our rebuttal?**
>
> Dear Reviewer NSUP, thanks for your time reviewing our paper. We have meticulously prepared a detailed response addressing your concerns and revised our paper accordingly. Could you please have a look to see if there are further questions? Your invaluable input is greatly appreciated. Thank you once again, and we hope you have a wonderful day!

---

### Official Review · Reviewer_fn47 · 2023-11-01

**Soundness:** 3 good
**Presentation:** 2 fair
**Contribution:** 3 good
**Rating:** 6
**Confidence:** 4

**Summary:**

The paper addresses the problem of robust overfitting in adversarial training (AT), where during adversarial training, the test robust accuracy of a model gradually decreases after a certain epoch (checkpoint) although the training robust accuracy continues to increase. This leads to a large robust generalization gap. The paper first provides a novel explanation for this phenomenon from the perspective of feature attribution. They divide the features learned by a model (robust DNN classifier) into `class-specific or exclusive features` and `cross-class features`. As their names suggest, class-specific features are informative for the prediction of a specific class, whereas cross-class features are informative for the prediction of multiple classes.

They provide empirical evidence to show that models trained using AT tend to rely more on cross-class features when observed at the best checkpoint (epoch where the test robust accuracy is highest). However, when robust overfitting occurs at later epochs/checkpoints, they observe that the model tends to rely less on cross-class features and more on class-specific features. In order to quantify the cross-class feature usage, they propose a feature attribution vector (per sample and per class) and a feature attribution correlation matrix. The feature attribution correlation matrix helps visualize the extent to which a model depends on cross-class features, and it is summarized by a metric named class attribution similarity (CAS). By observing the correlation matrix and CAS metric and at the best checkpoint and the final checkpoint across a number of datasets, architectures, and norm types, they conclude that there is consistently a higher dependence on cross-class features at the best checkpoint, but this dependence decreases at the final checkpoint, leading to robust overfitting. Therefore, it is crucial to preserve the model’s dependence on cross-class features to mitigate this phenomenon.

They provide a theoretical analysis to back this observation, based on a simple linear model with Gaussian class-conditional distributions and decoupled class-specific and cross-class features. Finally, they show that knowledge distillation can be effective at preserving the cross-class features. Therefore, they propose a knowledge distillation based adversarial training method where a weight-averaged model acts as the teacher for knowledge distillation. This method is shown to have improved adversarial robustness in their experiments.

**Strengths:**

1. The explanation for robust overfitting based on cross-class and class-specific features is intuitively clear and is backed well by empirical results.

2. The visualization of robust overfitting using the feature attribution correlation matrix and the CAS metric are useful tools for understanding this phenomenon. Sections 3.2 and 3.3 do a good job of considering multiple scenarios to understand the effect of cross-class features, and have pointers to the Appendix for additional results.

3. The weight average guided knowledge distillation method for adversarial training mitigates the issue  of robust overfitting and has improved performance on multiple datasets and architectures.

**Weaknesses:**

1. The theoretical analysis section is weak for the following reasons: lack of clarity in the presentation and theorem statements. The synthetic model is quite simple and can be presented in a better way (see my comments in the `Questions` section).

2. The proposed weight average guided knowledge distillation method (WAKE) is not clearly described. Is the weight-averaged model $\bar{\theta}$ created on the fly during adversarial training, similar to the self ensemble adversarial training (SEAT) method of (Wang & Wang, 2022)? The piecewise linear scheduling for $\lambda$ should be described precisely. More clarity is needed on these aspects, and an algorithm block would also be helpful (appendix can be used if space is limited).

3. Minor: experiments in the main paper are limited, and only two baselines are used for comparison. However, there are more experiments and other baselines like TRADES in the appendix.

4. Code has not been made available.

Would consider raising my score if Sections 4 and 5 are improved.

**UPDATE:** Increased my score to 6 after reading the author responses and revised paper.

**Questions:**

1. In Eqn (3), the max over the perturbation $\delta$ should include all three loss terms. It seems like the max is only over the cross-entropy loss term. Please make it clear by moving the $(1 - \lambda_1 - \lambda_2)$ term inside the max and using parentheses around the three loss terms.

2. The terms $\ell_{CE}$ and $\mathcal{KD}$ in Eqn (3) should be defined as the cross entropy loss and the Kullback-Leibler divergence respectively.

3. Nit: the $\times$ operator is not needed in Eqn (4).

4. First paragraph of section 3.1:
    - The notation $f(\cdot) = W \circ g(\cdot)$ seems incorrect. It should be $f(\cdot) = W g(\cdot)$, i.e. a matrix multiplication.
    - Please define the notation that $W[i]^T$ is the $i$-th row of $W$.
    - The dot is not needed for inner products and scalar products. For instance, it can be $f(x)_i = W[i]^T g(x)$ rather than $f(x)_i = W[i]^T \cdot g(x)$. Similarly, it can be $g(x)_j \\,W[i, j]$ rather than $g(x)_j \cdot W[i, j]$.

5. Referring to the subsection `Knowledge distillation mitigates robust overfitting`, the following statement is not clear: “. . . using knowledge distillation can preserve the weight of these features by smoothing the labels of the overlapping classes”.

### On Section 4
The description of the synthetic model is not clear in Section 4.1. Some suggestions below:

6. Use of $i$ to index the class is confusing. Instead, $y$ could be used to denote the class and $i$ or $j$ to index the features.

7. It is mentioned that the data distribution $\mathcal{D}_i = \\{ x\_{Ej}, x\_{Cj} \\,|\\, 1 \leq j \leq 3 \\} \in \mathrm{R}^6$. How can the data distribution be a set of features which lives in $\mathrm{R}^6$? I can follow what is implied, but it is not formally correct.

8. In Eqn (7), the probability distributions are actually **class-conditional** rather than marginal. That is, for the exclusive features: $x\_{Ej} \\,|\\, y = i \sim \mathcal{N}(\mu, \sigma^2)$ when $j = i$ and $0$ when $j \neq i$. Similarly, for the cross-class features: $x\_{Cj} \\,|\\, y = i \sim \mathcal{N}(\mu, \sigma^2)$ when $j \neq i$ and $0$ when $j = i$. Specifying it this way makes it clear why these features are defined as exclusive and cross-class.

9. In Eqn (9), it would be better to say the outer expectation is over the class prior distribution since $\mathrm{E}_i[\cdot]$ is pretty vague.

10. In Eqn (9), it is not clear how the cross-entropy loss simplifies to the term $\max_{j \neq i} f_w(x + \delta)_j - f_w(x + \delta)_i$. It seems. to me that if we simplify the cross-entropy loss, it reduces to $\log(\sum_j e^{f_w(x + \delta)_j}) - f_w(x + \delta)_i$. Same question applies to Eqn (10).

11. What is the significance of introducing the regularization term in this objective?

12. In Section 4.2, 2nd line: should it be `abandon $x_C$` rather than `abandon $x_E$`?

13. The statements of theorem 2 and theorem 3 are vague. They could be stated more formally. In theorem 3, what is $\epsilon_0$?

14. In the subsection `Knowledge distillation preserves cross-class features`, could the authors clarify how label smoothing applies here due to symmetry? And what does symmetry refer to here?

### On Section 5
15. As pointed out in point 2 under `Weaknesses`:
Is the weight averaged model $\bar{\theta}$ created on the fly during adversarial training, similar to the self ensemble adversarial training (SEAT) method of (Wang & Wang, 2022)? The piecewise linear scheduling for $\lambda$ should be described precisely. More clarity is needed on these aspects, and an algorithm block would also be helpful (appendix can be used if space is limited).

16. In Eqn (11), please clarify that the max over $\delta$ applies to both the loss function terms.

17. What is the temperature parameter $T = 2$ referred to in section 5.2 under settings?

18. Finally, the number of references seems to be small given the breadth of literature in this area.

---

> ### Author Response · Authors · 2023-11-20
> **Response to Reviewer fn47 (1/3)**
>
> Thank you for your detailed and valuable feedback. We have revised our manuscript based on your comments and address your concerns as follows.
>
> ---
>
> **Q1**: In Eqn (3), the max over the perturbation $\delta$ should include all three loss terms. It seems like the max is only over the cross-entropy loss term. Please make it clear by moving the $(1-\lambda_1-\lambda_2)$ term inside the max and using parentheses around the three loss terms.
>
> **A1**: Thanks for your careful reading. We have fixed this typo in Equation (3) in our manuscript.
>
> ---
>
> **Q2**: The terms  $\ell_{CE}$ and $\mathcal{KD}$ in Eqn (3) should be defined as the cross entropy loss and the Kullback-Leibler divergence respectively.
>
> **A2**: Thank you for your valuable comment. We'd like to clarify that $\mathcal {KD}$ is not the Kullback-Leibler divergence; rather, it represents the knowledge distillation function. This function initially computes the distilled probability prediction for both the student and teacher models by dividing the temperature $T$. Subsequently, it calculates the KL-divergence between these predictions:
>
> $\mathcal{KD}(f(\boldsymbol\theta_{\text{student}};x), f(\boldsymbol\theta_{\text{teacher}};x)) = \text{KL}[\text{softmax}(\frac{f(\boldsymbol\theta_{\text{student}};x)}{T}),
> \text{softmax}(\frac{f(\boldsymbol\theta_{\text{teacher}};x)}{T}$)]
>
> where $\text{KL}(\cdot,\cdot)$ is the Kullback-Leibler divergence and $T$ is the distillation temperature. We have clarified this in page 3 in our revision and added the details in Appendix G.2.
>
> ---
>
> **Q3**: Nit: the $\times$ operator is not needed in Eqn (4).
>
> **A3**: Thanks for pointing that out. We have removed that $\times$ operator in Eqn (4) to improve readability.
>
> ---
>
> **Q4**: First paragraph of section 3.1: Notations for $f(\cdot), W$ and $f(x)_j$.
>
> **A4**: Thank you for providing such detailed feedback. Based on your suggestions, we have addressed the typos in Section 3.1 on page 4 to enhance the overall readability of the manuscript.
>
> ---
>
> **Q5**: Referring to the subsection ``Knowledge distillation mitigates robust overfitting``, the following statement is not clear: “. . . using knowledge distillation can preserve the weight of these features by smoothing the labels of the overlapping classes”.
>
> **A5:** Thanks for your careful reading. In order to provide further clarity regarding the effectiveness of knowledge distillation for preserving cross-class features, we have refined the claim in our revised paper as follows:
>
> > In the process of AT with knowledge distillation, the teacher model adeptly captures the cross-class features present in the training data, and provides more precise labels by considering both class-specific and cross-class features. This stands in contrast to vanilla AT with one-hot labels, which primarily emphasizes class-specific features and may inadvertently suppress cross-class features in the model weights. The incorporation of cross-class features, backed by both our empirical findings and theoretical insights highlighting their significance for enhanced robustness, enables knowledge distillation to effectively mitigate robust overfitting by preserving these crucial features.
>
> ---
>
> **Q6**: Use of $i$ to index the class is confusing. Instead, $y$ could be used to denote the class and or to index the features.
>
> **A6**: Thanks for your kind consideration. Following your suggestion, we have replaced the index $i$ for classes with $y_i$ to differentiate them from the index of features in the revised paper.
>
> ---
>
> **Q7**: It is mentioned that the data distribution $\mathcal{D}\_i = \\{ x\_{E_j}, x\_{C\_j} | 1 \leq j \leq 3 \\} \in \mathrm{R}^6$. How can the data distribution be a set of features which lives in $R^6$? I can follow what is implied, but it is not formally correct.
>
> **A7**: Thanks for your careful checking. In Section 4.1, page 7 of our revised paper, we redefined the features for each sample as living in $\mathbb R^6$, and defined the distribution for the features independently in equation (7).
>
> ---
>
> **Q8**: In Eqn (7), the probability distributions are actually **class-conditional** rather than marginal. That is, for the exclusive features: $x_{Ej} \|\ y = i \sim \mathcal{N}(\mu, \sigma^2)$ when $j=i$ and 0 when $j\ne i$. Similarly, for the cross-class features: $x_{Cj} \|\ y = i \sim \mathcal{N}(\mu, \sigma^2)$ when $j\ne i$ and 0 when $j=i$. Specifying it this way makes it clear why these features are defined as exclusive and cross-class.
>
> **A8**: Thank you for your kind suggestion. We have revised the definition in equation (7) as per your suggestion to make it clearer.

---

> ### Author Response · Authors · 2023-11-20
> **Response to Reviewer fn47 (2/3)**
>
> **Q9**: In Eqn (9), it would be better to say the outer expectation is over the class prior distribution since $\mathrm{E}_i[\cdot]$ is pretty vague.
>
> **A9**: Thanks for pointing this out. Following your suggestion, we have modified $\mathbb E_i$ as $\mathbb E_{y_i\sim\{y_1,y_2,y_3\}}$ in both equations (9) and (10) to clarify the class prior for the expectation.
>
> ---
>
> **Q10**: In Eqn (9), it is not clear how the cross-entropy loss simplifies to the term $\max_{j \neq i} f_w(x + \delta)_j - f_w(x + \delta)_i$.
>
> **A10**: Thanks for your careful consideration, but we didn't claim that we apply cross-entropy loss in this framework, which may be hard to analyze. Instead, we use this simple loss function: $\max_{j\ne i} [f_w(x+\delta)\_j]-f_w(x+\delta)\_{y_i}$, where $y_i$ is the index of the true class. Both cross-entropy and this loss function encourage the model to increase the logit margin between the correct class $f_w(x+\delta)_{y_i}$ and any other class $f_w(x+\delta)_j\quad (j\ne y_i)$.
>
> ---
>
> **Q11**: What is the significance of introducing the regularization term in this objective?
>
> **A11**: Thanks for your careful reading, we explain the significance of this regularization term as follows. Adding this regularization term prevents both $w_1$ and $w_2$ from going infinitely large. For example, by multiplying a positive term $k>0$ to both $w_1$ and $w_2$, the weights become $kw_1$ and $kw_2$, and these parameters can also cause the loss function to be multiplied by $k$ due to the linearity of $\mathcal L$, but this will not change the classification results due to the linearity of $f_w$. Therefore, for any $w_1$ and $w_2$ that imply a negative loss function $\mathcal L(f_{w_1,w_2})$, multiplying $k>1$ makes $\mathcal L(f_{kw_1,kw_2})<\mathcal L(f_{w_1,w_2})$. Thus, there is no optimal solution for $\min \mathcal L(f_w)$ without a regularization term.
>
> ---
>
> **Q12**: In Section 4.2, 2nd line: should it be ``abandon $x_C$`` rather than ``abandon $x_E$``?
>
> **A12**: Thank you for your careful reading. It is exactly ``abandon $x_C$`` rather than ``abandon $x_E$``, and we have fixed this typo in our revision.
>
> ---
>
> **Q13**: The statements of theorem 2 and theorem 3 are vague. They could be stated more formally.
>
> **A13**: Thank you for your valuable comment, we have clarified the possibly confusing part in the statements in our revised paper and explained them as follows.
>
> - We have revised the claim in Theorem 2 more clearly as
>
>     > For any class $y$, consider weights $w_1 > 0$, $w_2 \in [0, w_1]$, and $\epsilon \in (0, \frac{\mu}{2})$. When sampling $x$ from the distribution of class $y$, increasing the value of $w_2$ enhances the possibility of the model assigning a higher logit to class $y$ than to any other classes $y'\ne y$ under adversarial attack. In other words, the probability $\Pr_{x\sim\mathcal D_y}[f_w(x+\delta))\_{y}>f_w(x+\delta)\_{y'},\forall \delta:\|\delta\|_\infty\le\epsilon]$ monotonically increases with $w_2$ within the range $[0, w_1]$.
>
> - For $\epsilon_0$ in Theorem 3, it is exactly the $\epsilon_0$ derived in Theorem 1. We have added this explanation in our revision.
>
> ---
>
> **Q14**: In the subsection ``Knowledge distillation preserves cross-class features``, could the authors clarify how label smoothing applies here due to symmetry? And what does symmetry refer to here?
>
> **A14**: Thank you for your careful reading. In this context, the term 'symmetry' specifically refers to the symmetry of logits for the other two classes when taking the expectation in the loss function (equation 10). When considering data from class $y$, both the distribution of features $x_{E,i}$  and $x_{C_i}$ for the other two classes, as well as their respective weights $w_1$ and $w_2$, exhibit symmetry respectively. Consequently, after applying knowledge distillation, the expectation for logits of the other two classes in the objective loss function (equation 10) becomes identical. To simplify this process, we can employ label smoothing.
>
> We have incorporated this discussion into our revision.

---

> ### Author Response · Authors · 2023-11-20
> **Response to Reviewer fn47 (3/3)**
>
> **Q15**: As pointed out in point 2 under Weaknesses: Is the weight averaged model created on the fly during adversarial training, similar to the self ensemble adversarial training (SEAT) method of (Wang & Wang, 2022)? The piecewise linear scheduling should be described precisely. More clarity is needed on these aspects, and an algorithm block would also be helpful (appendix can be used if space is limited).
>
> **A15**: Thank you for bringing this to our attention. The weight-averaged model, denoted as $\bar\theta$, is dynamically constructed on-the-fly and continues until the warm-up procedure concludes.
>
> During the warm-up stage of the weight-averaged model, the piecewise linear schedule for $\lambda$ is defined as follows: $\lambda$ starts at 0 and linearly increases to 1 until a certain epoch. Subsequently, $\lambda$ remains constant at 1 until the end of the training process.
>
> In response to your suggestion, we have incorporated a detailed algorithm outlining our proposed method in **Appendix G.1.**
>
> ---
>
> **Q16**: In Eqn (11), please clarify that the max over $\delta$ applies to both the loss function terms.
>
> **A16**: Thanks for your careful reading. We have corrected this in our revision.
>
> ---
>
> **Q17:** What is the temperature parameter $T$ referred to in section 5.2 under settings?
>
> **A17**: Thanks for your kind consideration, $T$ is the temperature for the knowledge distillation function. We have added more details on knowledge distillation in **Appendix G.2**, as discussed in our response in **A2**.
>
> ---
>
> **Q18**. Finally, the number of references seems to be small given the breadth of literature in this area.
>
> **A18**. Thanks for the kind advice. We have added additional related work in Appendix H.
>
> ---
>
> ### Minor comments
>
> **W3**: Experiments in the main paper are limited, and only two baselines are used for comparison. However, there are more experiments and other baselines like TRADES in the appendix.
>
> **A**: Thank you for recognizing that more comprehensive experiments are included in the appendix, where experiments over $\ell_2$-norm AT, more model architectures, and combinations with TRADES are included. As the proposed knowledge distillation method is designed to mitigate robust overfitting, to the best of our knowledge, KDSWA is the **only** existing knowledge distillation method for the same purpose. Thus, we only consider vanilla training and KDSWA as baselines.
>
> ---
>
> **W4**: Code has not been made available.
>
> **A**: We commit that we will release our code upon publication.
>
> ---
>
> We truly appreciate your valuable and detailed feedback. If you have any further questions or concerns, please let us know.

---

> ### Author Response · Authors · 2023-11-22
> **Could you please have a look at our rebuttal?**
>
> Dear Reviewer fn47, thanks for your time reviewing our paper. We have meticulously prepared a detailed response addressing your concerns and revised our paper accordingly. Could you please have a look to see if there are further questions? Your invaluable input is greatly appreciated. Thank you once again, and we hope you have a wonderful day!

---

> > ### Comment · Reviewer_fn47 · 2023-11-23
> >
> > Dear authors,
> >
> > Thank you for the detailed responses and revision to the paper, including the addition of Algorithm 2 for WAKE. Most of my concerns have been clarified. However, some of the changes made to the equations are not consistent with my suggestions. Please see below and address them in the next revision.
> >
> > Overall, I found the paper to make an interesting contribution by providing a novel understanding and knowledge distillation-based training approach for the robust overfitting problem. Hence, I will update my rating to 6.

---

> ### Comment · Reviewer_fn47 · 2023-11-23
> **Suggested changes (part 1)**
>
> Regarding Equations 3, 11, and 10, I meant changing them as follows because even in the revised version, it is not clear that the inner `max` over $\delta$ applies to all three terms.
>
> **Equation 3**\
> $\min_{\theta} \mathbb{E}_{(x,y) \sim \mathcal{D}\_{train}} \left[  \max\limits\_{\\|\delta\\|_p \le \epsilon} \tilde{\ell}(\theta \~;\~ \theta_1, \theta_2, x + \delta, y) \right] $, where \
> $\tilde{\ell}(\theta \~;\~ \theta_1, \theta_2, x + \delta, y) \~=\~ (1 - \lambda_1 - \lambda_2) \ell\_{CE}(f(\theta, x + \delta), y) \~+\~ \sum\limits\_{i=1}^2 \lambda_i \\, \mathcal{KD}( f(\theta, x + \delta), f(\theta_i, x + \delta) )$
>
> **Equation 11**\
> $ \max\limits\_{\\|\delta\\|_p \le \epsilon} \tilde{\ell}(\theta \~;\~ \bar{\theta}, x + \delta, y) $, where \
> $\tilde{\ell}(\theta \~;\~ \bar{\theta}, x + \delta, y) = (1 - \lambda) \ell\_{CE}( f(\theta, x + \delta), y ) \~+\~  \lambda \\,\mathcal{KD}( f(\theta, x + \delta), f(\bar{\theta}, x + \delta) )$
>
> Similar suggestion for **Equation 10** as well.

---

> > ### Author Response · Authors · 2023-11-23
> > **Further response to reviewer fn47**
> >
> > Dear reviewer fn47,
> >
> > Thank you so much for your prompt and valuable feedback. We are incredibly grateful for your recognition that our paper offers a novel perspective on robust overfitting and a training approach using knowledge distillation to tackle this problem. We are also very grateful for the increase in your score. Your approval is truly motivating and inspiring, and your kindness is sincerely appreciated.
> >
> > Furthermore, we have made revisions to our paper based on your suggestions regarding Equations 3, 9, 10, and 11. Your detailed suggestions have been invaluable in improving our paper. Thank you once again for your kind suggestions and comments. If you have any further concerns or questions, please let us know. We would be delighted to discuss them with you.

---

> ### Comment · Reviewer_fn47 · 2023-11-23
> **Suggested changes (part 2)**
>
> In the earlier version, the class label was denoted by $y = i$, which I found to be more clear than the current version (which denotes the label by $y_i$). For instance, the notation $x_{E_i}$ is clearer than $x_{E_{y_i}}$, and $f_w(x)\_i$ is clearer than $f_w(x)\_{y_i}$.
>
> As suggested in my original review, Eqn 7 is perhaps more clear when expressed as follows. The conditional distribution $\mathcal{D}\_i$ for class $i$ is defined as: \
> $x_{E_j} \~|\~ y = i \~\sim \begin{cases} \mathcal{N}(\mu, \sigma^2) \~\~&\text{ if  } j = i  \\\\  0 \~\text{ w.p. } 1  \~\~&\text{ if  } j \neq i \end{cases}$
>
> $x_{C_j} \~|\~ y = i \~\sim \begin{cases} \mathcal{N}(\mu, \sigma^2) \~\~&\text{ if  } j \neq i  \\\\  0 \~\text{ w.p. } 1  \~\~&\text{ if  } j = i \end{cases}$ \
> where $j \in \\{1, 2, 3\\}$.
>
> ### Regarding Equation 9
> My suggestion in the original review is a simple change, to indicate that the outer expectation is over the prior distribution on class labels $p_Y$, i.e.\
> $\mathcal{L}(f_w) = \mathbb{E}\_{y \sim p_Y}[ \mathbb{E}\_{x \sim \mathcal{D}_y}[  \cdots ] ] + \frac{\lambda}{2} \\|w\\|^2$
>
> That's all the changes I want to suggest. Glad to know that you found my feedback useful. Good luck!

---

> > ### Author Response · Authors · 2023-11-23
> > **Thanks again for your suggestions**
> >
> > Dear reviewer fn47,
> >
> > Thank you once again for your careful and detailed suggestions. We have further revised our paper accordingly. We truly appreciate your efforts in helping us polish our paper and make it above the standards of ICLR. We are also grateful for your recognition of the quality of our paper. Your approval is truly motivating and inspiring, and your kindness is sincerely appreciated. We sincerely thank you once again and wish you all the best.

---

### Meta-Review · Area_Chair_8Gbf · 2023-12-15

**Metareview:**

The paper presents a pretty interesting hypothesis of cross-class features and neural networks use different kinds of features over epochs when performing adversarial training. Unfortunately, many of the claims are unsubstantiated, and the new proposed methods do not match the intuition provided by the hypothesis. There are several such concerns raised by multiple reviewers. In my own read of the paper, I had the following big questions (seconded by reviewers):

I read the paper and could not understand how the authors explain why knowledge distillation helps (or with weight averaging). It feels almost tautological that a better teacher would help - it's unclear whether being better in preserving "cross-class" features is really the key here? Furthermore, the intuition of weight averaging helping also seems pretty straightforward and unrelated to the intuition described. Do you have any clarity here? (2) Reviewer NUSP raised a good point about validity of "class-specific" or "cross-class" features when taking average after perturbing points to different classes. Do you share a similar concern? (3) Reviewer QUY6 asks what's specific about adversarial training?

**Justification For Why Not Higher Score:**

See issues raised above regarding soundness of claims. The empirical gains are anyways small with limited novelty of proposed approach. The authors claim main point of the paper is the new insights but they are not clearly explained or substantiated.

**Justification For Why Not Lower Score:**

N/A

---

### Decision · Program_Chairs · 2024-01-16

Reject